# Altered Monocyte and Lymphocyte Phenotypes Associated with Pathogenesis and Clinical Efficacy of Progestogen Therapy for Peritoneal Endometriosis in Adolescents

**DOI:** 10.3390/cells13141187

**Published:** 2024-07-12

**Authors:** Elena P. Khashchenko, Lyubov V. Krechetova, Polina A. Vishnyakova, Timur Kh. Fatkhudinov, Eugeniya V. Inviyaeva, Valentina V. Vtorushina, Elena A. Gantsova, Viktoriia V. Kiseleva, Anastasiya S. Poltavets, Andrey V. Elchaninov, Elena V. Uvarova, Vladimir D. Chuprynin, Gennady T. Sukhikh

**Affiliations:** 1FSBI “National Medical Research Center for Obstetrics, Gynecology and Perinatology Named after Academician V.I. Kulakov” Ministry of Healthcare of the Russian Federation, 4, Oparina Street, 117997 Moscow, Russia; l_krechetova@oparina4.ru (L.V.K.); p_vishnyakova@oparina4.ru (P.A.V.); t_fatkhudinov@oparina4.ru (T.K.F.); e_inviyaeva@oparina4.ru (E.V.I.); v_vtorushina@oparina4.ru (V.V.V.); v_kiseleva@oparina4.ru (V.V.K.); a_poltavets@oparina4.ru (A.S.P.); a_elchaninov@oparina4.ru (A.V.E.); e_uvarova@oparina4.ru (E.V.U.); v_chuprynin@oparina4.ru (V.D.C.);; 2Research Institute of Molecular and Cellular Medicine, Peoples’ Friendship University of Russia (RUDN University), 117997 Moscow, Russia; gantsova_ea@pfur.ru; 3Avtsyn Research Institute of Human Morphology of Federal State Budgetary Scientific Institution “Petrovsky National Research Centre of Surgery”, 117418 Moscow, Russia; 4Department of Histology, Embryology and Cytology, Pirogov Russian National Research Medical University (Pirogov Medical University), 117997 Moscow, Russia; 5Department of Obstetrics and Gynecology, Sechenov First Moscow State Medical University, Trubetskaya str. 8, bld. 2, 119991 Moscow, Russia

**Keywords:** endometriosis, pelvic pain, macrophage, lymphocyte, CD206, CD163, CD86, CD56, CD16, NK cells, sVEGFR, adolescents

## Abstract

**Background**: Immunological imbalances characteristic of endometriosis may develop as early as the primary manifestations of the disease in adolescence. **Objective**: To evaluate subpopulation dynamics of monocytes and lymphocytes in peripheral blood and peritoneal fluid of adolescents with peritoneal endometriosis at diagnosis and after 1-year progestogen therapy. **Methods**: This study included 70 girls, 13–17 years old, diagnosed laparoscopically with peritoneal endometriosis (*n* = 50, main group) or paramesonephric cysts (*n* = 20, comparison group). Phenotypes of monocytes and lymphocytes of the blood and macrophages of the peritoneal fluid were analyzed by flow cytometry at diagnosis and during progestogen therapy. **Results**: Differential blood counts of CD16^+^ (*p* < 0.001) and CD86^+^ (*p* = 0.017) monocytes were identified as independent risk factors for peritoneal endometriosis in adolescents. During the treatment, cytotoxic lymphocytes CD56dimCD16bright (*p* = 0.049) and CD206+ monocytes (*p* < 0.001) significantly increased while CD163+ monocytes decreased in number (*p* = 0.017). The CD56dimCD16bright blood counts before (*p* < 0.001) and during progestogen therapy (*p* = 0.006), as well as CD206^+^ blood counts during the treatment (*p* = 0.038), were associated with the efficacy of pain relief after 1-year progestogen therapy. **Conclusions**: Adolescents with peritoneal endometriosis have altered counts of pro- and anti-inflammatory monocytes and lymphocytes both before and after 1-year progestogen therapy, correlating with treatment efficacy and justifying long-term hormonal therapy.

## 1. Introduction

The first symptoms of endometriosis often develop in adolescence, an average 7–10 years before diagnosis [1,2,3]. This delayed diagnosis and treatment may cause uncontrolled progression, undermine conservative treatment options and ultimately lead to infertility and impaired quality of life in young patients [4,5,6]. Despite the prevalence of endometriosis, particularly its peritoneal form (PE), clinical findings concerning initial forms of the disease in adolescents remain limited [7,8,9].

The pathogenesis of PE bears certain resemblance to neoplastic disorders; its features include proliferative growth of lesions, metabolic reprogramming of cells at the sites of engraftment in the peritoneum and immune reactions to the non-conventional composition of the cellular elements occupying peritoneal niches [7,10,11,12,13,14,15]. Particular attention is currently focused on pathogenetic treatment options for PE and corresponding markers of clinical efficacy [15,16,17]. Considering the early manifestation of the disease and actual need for a timely start to treatment, it is relevant to specifically focus on the pathogenetic mechanisms of PE in adolescents. In particular, characteristic alterations of molecular and cellular profiles, which are described for adults with PE, may emerge at early stages of pathogenesis in adolescents.

The progression of the peritoneal endometrioid foci is largely determined by the microenvironment at the sites of engraftment. Adult patients with PE show a local deficit of NK cell immunity associated with increased peritoneal fluid secretion volume [18] and increased counts of peritoneal macrophages [19]. Several studies show that polarized peritoneal macrophages with anti-inflammatory properties can support PE progression by stimulating proliferation of both epithelial and stromal cell lineages in the ectopic foci [20,21,22]. Colonization of the foci by macrophages under hyperestrogenic conditions facilitates the local production of neuroactive peptides and growth factors that support neoangio- and neurogenesis, notably Sema3A, NGF and VEGF [23]. Specification of molecular and cellular parameters of the course of PE in adolescents at diagnosis and during therapy is highly relevant in terms of clinical management and advanced treatment options.

In this study, we will address the subpopulation dynamics of monocytes and lymphocytes in the peripheral blood and peritoneal fluid of adolescents with PE at diagnosis and after 1-year progestogen therapy.

## 2. Methods

We performed a case–control study which included 50 girls from menarche to 17 years old (16.0 (15.0–17.0)) with laparoscopically confirmed diagnosis of peritoneal endometriosis (PE) (*n* = 50 in the main group) and paramesonephric cysts (*n* = 20 in the comparison group). All adolescents underwent inpatient treatment in 2020–2022 in the Department of Pediatric and Adolescent Gynecology at the V.I. Kulakov National Medical Research Center for Obstetrics, Gynecology and Perinatology, Moscow, Russia.

The inclusion criteria in the main group were the following: (1) age of girls from menarche to 17 years old inclusively; (2) laparoscopically confirmed diagnosis of peritoneal endometriosis; (3) absence of drug administration for at least 3 months preceding the study (psychotropic, any hormonal drugs, including combined oral contraceptives); and (4) informed consent obtained from the patient or her legal representative (for girls under 15 years old) for participation in the research.

The indications for operative surgery in the main group were a persistent moderate or severe degree of dysmenorrhea and/or chronic pelvic pain, resistant to empirical treatment with NSAIDs for at least 3 months preceding the study, with suspicion of genital endometriosis according to MRI.

The exclusion criteria for the main group were the following: age of patients over 18; aggravation of any chronic or acute diseases (infections, endocrine disorders, oncology, etc.); mental disorders; tumors of pelvic organs; absence of secondary dysmenorrhea or chronic pelvic pain; hereditary or congenital malformations, in particular associated with obstruction of menstrual outflow; and absence of informed consent.

The comparison group in the study consisted of 20 adolescents of the same age (16.0 (15.0–17.0)).

In the comparison group, we included girls with regular, unpainful, moderate periods (within 24–38 days, lasting 4–7 days) without any gynecological pathology except paramesonephric cyst (the diameter of the cyst was more than 4 cm and less than 6 cm). The girls also had no endocrine/somatic pathology and did not routinely take any drugs for at least 3 months preceding this study; informed consent was also obtained from the patient or her legal representative for participation in this study.

The exclusion criteria in the comparison group were almost the same as with the main group: age of patient over 18 years old; any chronic or acute disease; mental, endocrine, hereditary, congenital or gynecological diseases; tumors of the pelvic organs (except for paraovarian cyst, or its diameter being more than 6 cm); secondary dysmenorrhea or chronic pelvic pain; absence of informed consent for participation in the research. The indication for surgery in the comparison group was ultrasound (US) or MRI data of a paraovarian cyst.

### 2.1. Laparoscopy

This study included patients with PE confirmed by laparoscopic intervention, in the course of which the observable peritoneal lesions were (mostly) excised or coagulated. The laparoscopic report for each case included a surgical detailed diagnosis, the stage of the disease according to the revised American Society for Reproductive Medicine (rASRM classification, revised in 2011) criteria and the revised Enzian Classification (#Enzian, revised in 2021), and a description of the lesions in terms of localization, color, size and depth [24,25]. The histological data included gross and microscopic descriptions of the tissue.

During the laparoscopic intervention, tissue fragments and peritoneal fluid were obtained. In the control group, the peritoneum was biopsied in the area of the right lateral peritoneal recess, and in the main group, biopsies of endometrioid lesions in the pelvic peritoneum (mainly in the area of the ovarian fossa or pouch of Douglas) and intact peritoneum in the area of the right lateral recess were collected. The tissues were placed in liquid nitrogen for cryostorage at −80 °C immediately upon collection; thawing was carried out immediately before the analysis.

After laparoscopy, all patients of the main group received prolonged pathogenetic therapy with dienogest at a dose of 2 mg daily without interruptions. A year later, during treatment, the clinical picture and peripheral blood parameters were reanalyzed.

### 2.2. Isolation of Mononuclear Cells from Blood

To isolate peripheral blood mononuclear cells, 2 mL of whole blood-EDTA was collected in a tube and 8 mL of erythrocyte lysis buffer was added. The tube was kept at 4 °C for 10 min and centrifuged at 300× *g* at 4 °C for 5 min; the supernatant was removed and the pellet was washed in 10 mL of phosphate-buffered saline (PBS). After a second centrifugation, the supernatant was removed and the cells were fixed in 2% paraformaldehyde. On the day of analysis, the cells were washed, resuspended in PBS with 0.1% bovine serum albumin (BSA; Miltenyi Biotec, Bergisch Gladbach, Germany), counted, diluted to a density of 105 cells in 100 µL, incubated with fluorophore-labeled monoclonal antibodies at 4 °C for 15 min in the dark and washed with autoMACS^®^ Rinsing Solution (Miltenyi Biotec) at 500 g for 5 min; the pellets were resuspended in 400 µL PBS for cytometry.

All samples were analyzed in a FACSCalibur Flow Cytometer (Becton Dickinson, Franklin Lakes, NJ, USA). A minimum of 104 cells for each measurement were localized to a region of interest in a forward scatter vs. side scatter (FS–SS) plot comprising lymphocytes and monocytes. Samples without fluorophores were used as controls. The gating used pan-leukocyte marker CD45; an example is shown in Appendix A. The monocyte markers included conditionally M2 anti-inflammatory (CD206, CD163) and M1 pro-inflammatory (CD86, CD80) characteristics of monocytes and macrophages: CD206 (130-095-131; Miltenyi Biotec), CD86 (130-116-160; Miltenyi Biotec), CD16 (A07766; Beckman Coulter, Brea, CA, USA), CD80 (130-117-683; Miltenyi Biotec), CD163 (130-097-630; Miltenyi Biotec), CD56 (130-113-871; Miltenyi Biotec), CD192 (130-118-338; Miltenyi Biotec) and HLA-DR (130-111-789; Miltenyi Biotec). The flow cytometry data were analyzed using FlowJo LLC v10 software (FlowJo LLC, Ashland, OR, USA).

### 2.3. Isolation of Mononuclear Cells from Peritoneal Fluid

Each peritoneal fluid sample was centrifuged at 300× *g* at 4 °C for 20 min; the supernatant was discarded and the sample was further centrifuged in paraformaldehyde at 2000× *g* at 4 °C for 10 min. The pellet was mixed with 100 µL PBS and 900 µL erythrocyte lysis buffer. The tube was placed on ice for 10 min and centrifuged at 300× *g* at 4 °C for 5 min. The supernatant was removed and the pellet was resuspended in 10 mL PBS. After a second centrifugation, the pellet resuspended in 100 µL PBS was mixed with 100 µL of 4% paraformaldehyde, incubated for 10 min at room temperature, washed in PBS, kept at 4 °C and analyzed by flow cytometry as described in the previous paragraph; the markers included CD192 (130-118-338; Miltenyi Biotec), HLA-DR (130-111-789; Miltenyi Biotec), CD206 (130-095-131; Miltenyi Biotec), CD80 (130-117-683; Miltenyi Biotec), CD163 (130-097-630; Miltenyi Biotec), CD56 (130-113-871; Miltenyi Biotec), CD16 (A07766; Beckman Coulter) and CD86 (130-116-160; Miltenyi Biotec).

### 2.4. Lymphocyte Flow Cytometry and Functional Tests

Peripheral blood lymphocytes were phenotyped by flow cytometry using PerCP-, FITC- or PE-conjugated monoclonal antibodies (Becton Dickinson, USA) to CD3+ (T cells), CD3+ CD4+ (#342405, T helpers), CD3+ CD8+ (#342406, cytotoxic T cells), CD3− CD56+16+ (#342403, NK cells), CD3+ CD56+16+ (NKT cells), CD19+ (B cells) and CD19+ CD5+ (#333190, B1 cells). To identify individual subsets of NK cells in peripheral blood, a combination of CD16 (#555406) and CD56 (#345812) antibodies conjugated to different fluorophores was used. NK cells with high and low CD56 surface expression densities (CD56bright and CD56dim) differ by their functional properties: CD56brightCD16–/dim NK cells are considered efficient producers of cytokines with immunoregulatory properties but are capable of becoming cytotoxic upon activation, while CD56dim NK cells exhibit higher cytotoxicity by expressing more immunoglobulin-like receptors as well as Fcγ receptors (CD16).

Phagocytic activity of neutrophils (PAN) was assessed using the FagoFlow kit (Cat. #ED7042; ExBio, Praha, Czech Republic). The test is based on the assessment of the oxidative burst in granulocytes after stimulation with *E. coli*. The ratio of the average fluorescence intensity of activated granulocytes of stimulated samples and negative controls reflects the intensity of the oxidative burst of granulocytes after stimulation with *E. coli* and is designated as the stimulation index (SI).

Lymphocyte phenotyping and PAN measurements were performed in a Gallios flow cytometer (Beckman Coulter, Brea, CA, USA) using Kaluza Analysis Software 1.3.

### 2.5. Immunosorbent Assay

Concentrations of vascular endothelial growth factor A (VEGF-A), its soluble receptor 2 (sVEGFR2) and hypoxia-inducible factor 1α (HIF-1α) were measured by enzyme-linked immunosorbent assay using RayBiotech systems (Peachtree Corners, GA, USA), https://www.raybiotech.com/, accessed on 12 June 2024. The results were recorded using an Infinite F50 spectrophotometer (Tecan, Männedorf, Switzerland), https://lifesciences.tecan.com/products/microplate_readers/infinite_f50, accessed on 12 June 2024. Concentrations of the transforming growth factor TGF-β isoforms (TGF-β1, TGF-β2, TGF-β3) were measured in a Bio-Plex^®^ 200 System #171000201 (Bio-Plex Manager™ 6.0 software, Bio-Rad, Hercules, CA, USA) using Bio-Plex Pro TGF-β Panel 3-Plex immunoassay kits (Bio-Rad); the data were processed in the Bio-Plex Manager 6.0 Properties application (Bio-Rad).

**Statistical data analysis** was performed using Statistica 12 software (StatSoft Inc., USA) and IBM SPSS Statistics 27. Categorical variables were assessed by calculating frequencies and proportions (%) to compare differences, contingency tables were used and chi-squares (χ^2^) were calculated. For small samples, Fisher’s exact test or the χ^2^-Pearson test with Yates continuity correction was determined. Comparison of multiple frequencies was carried out by performing the χ^2^-Pearson test. McNemar’s chi-square was used to analyze frequency differences in the dependent sample before and after treatment.

The distribution normalities were challenged with distribution histograms visually and indicators of asymmetry, kurtosis, Kolmogorov–Smirnov and Shapiro–Wilk criteria. In addition to checking the type of distribution of variables, the homogeneity of variances in the study groups was assessed using analysis of variance methods and the Levene and Brown–Forsythe tests. Comparison of variables in two groups with a parametric distribution was analyzed with Student’s *t*-test (a mean value (M) and a standard deviation (SD) are presented). In the case of multiple groups, analysis of variance (ANOVA) was performed. Nonparametrically distributed variables were presented with median (Me) and interquartile range values and analyzed with the Mann–Whitney U-test. Comparison of nonparametric variables in multiple groups was performed with the use of the Kruskal–Wallis rank tests. Subsequently, the intergroup differences were assessed by post hoc testing with ranking according to the Dunn or Siegel–Castellan criterion. The adjusted odds ratio (OR) with a 95% confidence interval (CI) were evaluated using logistic regression methods to analyze the influence of various risk factors. The parameters in the dependent samples were assessed using the Wilcoxon signed rank test and χ^2^ McNemar’s test for dependent proportions; the differences were accepted as significant at *p* < 0.05. The data are presented as the median, 25–75% quartiles, min–max. Correlations were determined using Pearson’s correlation coefficient if the data were normally distributed or Spearman’s rank test for nonparametric data. The impact of categorical or quantitative factors were evaluated by factorial ANOVA and multiple logistic regression methods.

## 3. Results

### Clinical Characterization

According to the revised American Society for Reproductive Medicine (rASRM) classification, patients of the main group were distributed according to the following: 18.0% classified as stage I (9/50, the rASRM score 3.2 ± 1.3 AND P1-2, O 0/0, T0/0, B0-1/0-1 C0 #Enzian(s)); 36.0% classified as stage II (18/50 the rASRM score 10.5 ± 3.1 AND P1-2, O 0-1/0, T0-1/0, B1-2/1-2, C0 #Enzian(s)); and 46.0% classified as stage III (23/50, the rASRM score 23.4 ± 7.7 AND P2-3, B1/2 #Enzian(s)). Five cases classified as stage III rASRM (O2–3) presented with endometrioid ovarian cysts with adhesions formed between the ovary and the pelvic wall/uterosacral ligaments (T2–3).

Patients in the main and comparison groups did not differ significantly in anthropometric parameters and body mass index (20.9 ± 3.1 vs. 21.8 ± 5.4 kg/m^2^, *p* = 0.11).

Patients in the main and comparison groups did not differ significantly in age at menarche (12.1 ± 1.3 vs. 11.8 ± 1.4 years, *p* = 0.71). Patients with endometriosis were characterized in a third of cases by irregular (36.0%, 18/50 vs. 0%, 0/20, *p* = 0.002) and heavy menstruation (46.0%, 23/50 vs. 20.0%, 4/20, *p* = 0.044). The main complaint of patients with endometriosis was very severe (56.0%, 28/50, 9.21 ± 0.83, vs. 0%, 0/20, *p* < 0.001) and moderately severe dysmenorrhea (44.0%, 22/50, 6.14 ± 0.92, vs. 15.0%, 3/20, 3.26 ± 2.71, *p* = 0.002). In addition, girls with endometriosis were characterized by chronic pelvic pain (82.0%, 41/50, vs. 0%, 0/20, *p* < 0.001) and non-menstrual pain (32.0%, 16/50, vs. 0%, 0/20, *p* < 0.001), resistance to NSAIDs and antispasmodics (44.0%, 22/50, vs. 0%, 0/20, *p* < 0.001), and non-menstrual spotting (14.0%, 7/50 vs. 0%, 0/20, *p* < 0.001).

Comparison of general blood test parameters, biochemical and hormonal profiles, and blood coagulation system parameters did not reveal differences between the groups, with the exception of a higher level of prolactin in patients with endometriosis compared to the comparison group (468.21 ± 303.23 mlU/L vs. 276.52 ± 125.91 mlU/L, *p* = 0.041) and lower values of platelets (276.89 ± 64.14 10^9^ per L vs. 313.82 ± 85.14 10^9^ per L, *p* = 0.042) and thrombocrit (0.27 ± 0.05 L/L vs. 0.31 ± 0.07 L/L, *p* = 0.016) in patients with endometriosis. It is important to note that in adolescents, the level of AMH in the groups was also comparable (4.24 ± 2.55 ng/mL vs. 4.46 ± 2.72 ng/mL, *p* = 0.6). For the indirect markers of inflammation, C-reactive protein (1.03 ± 1.02 mg/L vs. 0.81 ± 0.79 mg/L, *p* = 0.4), the level of CA-125 (25.41 ± 25.93 U/mL vs. 15.96 ± 6.10 U/mL, *p* = 0.8), Ca-19.9 (18.82 ± 26.14 U/mL vs. 18.55 ± 17.49 U/mL, *p* = 0.6), HE4 (49.82 ± 9.51 pmol/L vs. 50.75 ± 12.24 pmol/L, *p* = 0.9), fibrinogen (2.65 ± 0.51 g/L vs. 2.77 ± 0.48 g/L, *p* = 0.4), glucose (4.68 ± 0.50 mmol/L vs. 4.78 ± 0.25 mmol/L, *p* = 0.4), total iron in the blood serum (Fe2+) (20.71 ± 9.74 µmol/L vs. 20.23 ± 9.38 µmol/L, *p* = 1.0) and ferritin (29.46 ± 21.94 µg/L vs. 28.50 ± 18.56 µg/L, *p* = 0.7) levels did not differ significantly between the two groups. Despite differences previously identified for adult patients with endometriosis compared with healthy controls [26], in our study, the ratios of blood cells (neutrophils to lymphocytes and platelets to lymphocytes) did not differ significantly between the groups. None of the indicators were identified as risk factors for the disease or attributed with diagnostic value by logistic regression and ROC analysis for adult cohorts [27].

Blood mononuclear cell and peritoneal fluid macrophage composition at diagnosis

Comparative analysis of monocyte subsets revealed lower differential counts of CD16+ peripheral blood monocytes and higher differential counts of peripheral blood monocytes that were positive for pro-inflammatory polarization marker CD86 and anti-inflammatory polarization marker CD206 (Table 1, Figure 1). All other markers of monocyte/macrophage subpopulations in the blood and peritoneal fluid did not differ significantly between the groups. It is noteworthy that the percentage of monocytes positive for the anti-inflammatory marker CD206 in peritoneal fluid did not differ significantly between groups.

Lymphocyte subpopulation analysis revealed no between-the-group differences except lower counts of B1 cells (CD19^+^CD5^+^) in the PE vs. comparison group at the time of laparoscopy.

The overall percentage of cells expressing the monocyte migration marker CD192 and the pro-inflammatory marker CD80 at the surface was higher in the PE vs. comparison group. The CD192^+^ and CD80^+^ differential cell counts showed strong positive correlation in both blood (r = 0.805, *p* < 0.001, Spearman’s rank test hereinafter) and peritoneal fluid (r = 0.844, *p* < 0.001). Furthermore, the CD192^+^ differential counts in peritoneal fluid positively correlated with serum levels of Hif-1α (r = 0.484, *p* = 0.036) and iron (r = 0.496, *p* = 0.031). Positive correlations were also found for pro-inflammatory activated monocyte subsets of the blood, HLA-DR^+^ и CD80^+^ (r = 0.453, *p* = 0.016) and CD86^+^ (r = 0.597, *p* = 0.005).

The percentage of peritoneal fluid macrophages expressing the anti-inflammatory marker CD163 negatively correlated with serum levels of Hif-1α (r = −0.357, *p* = 0.048), while the percentage of CD163^+^ cells among peritoneal fluid macrophages positively correlated with sVEGF-R2 levels (r = 0.354, *p* = 0.039), which may indirectly link the anti-inflammatory macrophage phenotypes to neoangiogenesis in PE.

In addition, the percentage of cells expressing pro-inflammatory marker CD86 positively correlated with CA-125 levels in blood (r = 0.394, *p* = 0.046) and in peritoneal fluid (r = 0.681, *p* = 0.044).

In addition to the comparative analysis of monocyte and lymphocyte subsets between the two groups, we stratified the main group (PE) into subgroups depending on the severity of the disease. For patients with stage I–II (27/50) and patients with stage III (23/50) PE, most of the studied laboratory parameters were similar. However, absolute counts of CD3+CD4+ lymphocytes (T helper cells) were significantly lower in patients with stage I–II as analyzed separately vs. the comparison group (0.71 (0.52–0.83) vs. 0.87 (0.78–0.94), *p* = 0.038). Of note are that the absolute counts of CD19^+^CD5^+^ lymphocytes (B1 cells) were significantly higher in the stage III subgroup compared to the stage I–II subgroup (0.039 (0.020–0.053) vs. 0.018 (0.013–0.033), *p* = 0.042); moreover, the CD19^+^CD5^+^ counts were the only significant factor for severity of the disease (Chi^2^ = 4.26, *p* = 0.039, OR 6.33, percent correct 74.19%) among the tested parameters.

The identified differences were further analyzed as putative risk factors for PE using logistic regression. The analysis identified decreased CD16+ monocyte blood counts as a significant risk factor for PE (Chi^2^ = 14.023, *p* = 0.0002), with an estimated accuracy of 85.0% (OR 35.00, percent correct 85.0%) and also increased CD86+ monocyte blood counts as a less accurate predictor (Chi^2^ = 5.67, *p* = 0.0171, OR 5.33; percent correct 70.00%).

The PE-associated opposite alterations in CD16+ and CD86+ monocyte counts (lower of CD16+ and higher of CD86+) were consistently observed for the entire main group (F = 6.84, *p* = 0.003) as well as in patients with persistent dysmenorrhea (F = 15.36, *p* < 0.001), patients with chronic pelvic pain (F = 14.36, *p* < 0.001) and patients with PE confirmed by histological examination (F = 5.442, *p* = 0.008) (Figure 2).

The analysis of soluble factor levels revealed decreased content of VEGF and its soluble receptors in the serum of patients with PE vs. the comparison group (Table 2, Figure 1). The VEGF-A level was identified as a significant risk factor for the disease (OR 1.23 (DI 1.01–1.49), *p* = 0.035), albeit diagnostically irrelevant due to its low classification significance.

Despite the lack of significant associations of TGF-β blood levels with PE, the analysis revealed significant correlations involving TGF-β isoforms. In particular, TGF-β1 levels positively correlated to differential and absolute blood counts of CD3^−^CD56,16^+^ NK cells (r = 0.447, *p* = 0.004 and r = 0.509, *p* = 0.005, respectively). Also, TGF-β3 levels positively correlated with CD55^bright^CD16^dim^ NK cell counts (r = 0.500, *p* = 0.029), while TGF-β3 levels negatively correlated with counts of monocytes positive for NK cell marker CD56 (r = −0.487, *p* = 0.035).

2.Clinical and laboratory characterization of patients with peritoneal endometriosis after 1-year progestogen therapy

During 1-year progestogen (dienogest, 2 mg) therapy, the patients noted a significant decrease in pelvic pain symptoms both during and outside menstruation, up to a complete elimination in 86.0% (43/50) of the cases. After 1 year of the therapy, examination recorded amenorrhea in 64.0% (32/50), rare menstruation to oligomenorrhea in 12.0% (6/50) and spotting episodes in 28.0% (15/50) of the cases. At the same time, 14.0% (7/50) of the main group showed a lack of dynamics during the therapy, with PE foci revealed by MRI.

Comparative analysis involved blood test data for 24 patients who were on dienogest for 12 months (Table 2).

During the therapy, the levels of estradiol and CA-125 decreased within reference ranges (Table 2), matching those in the comparison group. The decreasing levels of adrenal hormones (cortisol, 17-OHP and androstenedione) probably reflect the resolution of chronic pelvic pain as a stress factor. The increase in ferritin (bound iron) levels can be attributed to a reduced loss of iron due to suppressed menstrual bleeding in the majority of patients on dienogest. The lack of a significant decrease in AMH levels during the treatment is encouraging in terms of ovarian reserve maintenance.

3.Blood mononuclear cell composition in patients with peritoneal endometriosis after 1-year progestogen therapy

CD206+ monocyte counts increased significantly during the treatment of PE to levels significantly exceeding those in the comparison group. CD163+ monocyte counts, by contrast, significantly decreased during the treatment for PE to levels significantly lower than in the comparison group (Figure 3 and Table 3). In addition, the percentage of cytotoxic CD56dimCD16bright NK cells significantly increased during the treatment (Figure 3 and Table 3).

The neutrophil-to-lymphocyte ratio (NLR) was significantly decreased in patients with PE after the therapy (*p* = 0.008). Of note, NLR in patients with PE was statistically similar to the comparison group for the entire observation period.

Logistic regression analysis identified the percentage of cytotoxic CD56dimCD16bright cells both before treatment (OR 5.25, DI 0.44–62.12, *p* < 0.001) and after treatment (OR 1.92, DI 0.94–3.93, *p* = 0.006) as significant predictors of the pain relief efficacy. The accuracy of pain relief prediction using the CD56dimCD16bright cell counts before the therapy was 91.3% (Chi^2^ = 11.67, *p* < 0.001) (Figure 4).

At the same time, the cytotoxic CD56dimCD16bright NK cell counts were higher in patients with pain symptoms persisting (F = 9.81, *p* = 0.006) and NSAIDs demanded for pain relief (F = 6.59, *p* = 0.030) throughout the therapy (Figure 3). A similar trend was revealed for CD56brightCD16dim NK cells known to produce cytokines with immunoregulatory properties [28,29,30,31]. The persistence of the pain factor during therapy was positively associated with CD56brightCD16dim immunoregulatory (F = 8.99, *p* = 0.007) and CD56dimCD16bright cytotoxic (F = 7.88, *p* = 0.004) NK cell counts (Figure 3). A positive correlation of the absolute CD56brightCD16dim NK cell counts in the blood after treatment with the stage (r = 0.498, *p* = 0.022) and total rASRM scores for PE (r = 0.520, *p* = 0.016) was evident as well.

The treatment-related dynamics of CD206+ monocytes in the blood were also associated with pain relief efficacy (OR 0.82, DI 0.62–1.08, Chi^2^ = 4.29, *p* = 0.038), while CD163+ monocyte counts showed no significant association with the persistence of pain symptoms nor other clinical indicators of the treatment efficacy.

Of the pre-treatment indicators that differed significantly between the groups at the time of diagnosis, CD16^+^ monocyte counts were significantly associated with the resolution of pain symptoms by the treatment (OR 0.26, DI 0.003–20.12, Chi^2^ = 4.98, *p* = 0.026). It is interesting to note that along with an increase in CD16^+^ NK cell counts in patients with treatment-resistant pains, the CD16^+^ monocyte counts decreased significantly, and a similar trend was observed in patients with treatment-related amenorrhea (F = 5.577, *p* = 0.065).

According to factor analysis, none of the monocyte and lymphocyte subset counts were significantly associated with the recurrence of PE lesions according to MRI data collected after 12 months on progestogen treatment.

For progestogens in PE, treatment-related amenorrhea is considered a positive indicator of clinical efficacy. Significant reciprocal dynamics were present in mononuclear cells and CD206^+^ monocytes during therapy in the blood of patients with PE on dienogest presenting with treatment-related amenorrhea: a decrease in CD206^+^ monocytes and a parallel increase in mononuclear cell counts (Wilks’ lambda = 0.441, F = 10.76, *p* = 0.0009). However, the factor of amenorrhea development during treatment had no significant association with CD56dimCD16bright and CD56brightCD16dim lymphocyte counts.

## 4. Discussion

Peritoneal endometriosis is associated with profound immunological alterations and local immunosuppression [32,33,34]. In adolescents diagnosed with PE and treated for the condition, the immunological aspect had not been addressed previously, at least in the available published literature.

Immune cell pools of systemic circulation, peritoneal fluid, endometrioid lesions and eutopic endometrium are engaged in complex interactions, [14]. Various leukocytes, notably monocytes, release pro- and anti-inflammatory cytokines, chemokines and prostaglandins with multiple effects ranging from local vasoconstriction to neoangio- and neurogenesis, focal proliferation and dissemination of endometrioid heterotopias [16,35,36]. During the formation of endometrioid lesions, inflammatory cells are recruited to the lesions and produce inflammatory, nociception and growth factors including IL-1β, IL-37, IL-8, IL-12, IL-6, TNF-α, MCP1, glycodelin, NGF, PGE2, PGF2α, adhesion molecules ICAM and VCAM, metalloproteinases (MMPs-2, -3, -7, -9, -11), etc. [37,38]. The number of macrophages that activate T and B lymphocytes eventually increases, and the immune cell pools undergo compositional and functional alterations which ultimately mediate immunosuppression and favor the continued growth and progression of the disease. Recent findings identified endometriosis as an immune-dependent disease involving the violation of immunoregulatory mechanisms [39].

The concept of this study used data obtained for peritoneal endometriosis (PE) in adult patients, due to the apparent lack of clinical studies evaluating treatment efficacy for PE in adolescents on a comprehensive basis involving molecular and cellular characterization. The results confirmed the assumption that immune imbalances and their dynamics during therapy are typical already at the initial stages of PE in adolescents. However, due to the known age-related specifics of immune functionalities, data obtained both before and during therapy in adolescent patients with PE may differ from those for adult cohorts.

The analysis of monocyte phenotypes in adolescent patients with PE vs. the comparison group of conditionally healthy matching individuals revealed higher counts of monocytes positive for both anti-inflammatory (CD206) and pro-inflammatory (CD86) markers.

CD206, also known as macrophage mannose receptor (MMR), is thought to play a key role in the internalization of glycoproteins by concentrating antigens for subsequent transport to processing compartments of the cell. The mannose receptor, originally found on the surface of tissue macrophages, is expressed mainly on the surface of immature dendritic cells and some other cell types of mesenchymal origin, notably endothelial cells of lymph vessels, liver, spleen, etc. The mannose receptor is not expressed on the surface of lymphocytes and blood monocytes under normal conditions, but emerges on the surface of monocytes during their differentiation into macrophages, playing a role in the implementation of innate and acquired immunity. CD206 is a marker of conditional anti-inflammatory polarization, and its expression can be triggered by IL-4 and IL-13 [40]. A specific increase in blood counts of CD206^+^ monocytes in patients with PE indicates there was an imbalance of pro- and anti-inflammatory factors already at the initial stages of the disease. Polarized macrophages with an anti-inflammatory phenotype are involved in tissue remodeling and the secretion of immunomodulatory mediators including IL-10, TGF-β, IL-4 and IL-13 [40]. The pathogenetically relevant TGF-β pathway activated by CD206^+^ macrophages in PE [41] has been associated with the onset of hypoxia, cell metabolic reprogramming and transition to glycolysis as the dominant energy production mode, concomitant acidosis and angiogenesis in the foci [37,42,43].

CD206 and CD163 are scavenger receptors expressed by both monocytes and macrophages. CD163 expression can be induced by anti-inflammatory cytokines (IL-6, IL-10) and suppressed by pro-inflammatory stimuli (IL-4, TNF-α, IFN-γ, lipopolysaccharide) [44]. CD163 is involved primarily in the clearance of destroyed hemoglobin in complex with haptoglobin from circulation through endocytosis and subsequent heme metabolization by macrophages, as well as in the clearance of inflammatory molecules and regulation of apoptosis [45]. CD163+ blood monocyte counts in patients with PE were unaltered vs. the comparison group before the treatment but decreased during the treatment, possibly in connection with treatment-related secondary amenorrhea and the absence of retrograde menstruation. The decreasing counts of CD163+ blood monocytes during the treatment may indirectly indicate a decrease in oxidative stress levels within the peritoneal cavity due to lower rates of menstrual blood hemolysis and free iron levels, representing a prognostically favorable factor negatively associated with the growth of endometrial heterotopias in the peritoneum. The positive correlation between CD163+ monocyte counts in the peritoneal fluid and blood sVEGFR observed before the treatment supports the pathogenetic role of the imbalance of anti-inflammatory macrophages and the nutrition of the foci in the development of PE already at the initial stages.

Another studied marker, CD86, significantly increased in the blood of patients with endometriosis, is constitutively expressed on Langerhans cells, memory B cells, macrophages and monocytes. CD86 expression increases rapidly following activation by Ig receptor cross-linking or by addition of certain cytokines, for example, IFN-γ stimulation. Increased counts of CD86+ cells in patients with endometriosis, especially accompanied by chronic pelvic pain and persistent dysmenorrhea, indicates an extensive inflammatory process and systemic immune response [41].

Hai Zhu et al. (2022) [38] analyzed macrophage immunophenotypes in tissues of patients with endometriosis (aged 25–45 years) and supplemented the data with a study in an animal model. Our findings are consistent with the authors’ regarding an increase in the level of pro-inflammatory CD86^+^ macrophages in the endometrioid heterotopias of patients with pain compared to patients without pain. With the CD206 marker, the trends were somewhat opposite; however, we assessed the blood cell counts, while the authors demonstrated a significantly lower content of CD206^+^ cells in the foci of endometriosis in patients with pain. In addition, the authors showed that the polarization of macrophages, both CD86^+^ and CD206^+^, was accompanied by the formation of TRPV1/TRPA1 calcium channel heteromers in heterotopias and was associated with endometrial cell migration [38]. Formation of TRPV1/TRPA1 heteromeric calcium channels promotes the sensitization of peripheral nerve endings and enhances pain intensity in endometriosis. Izumi G. et al. (2017) demonstrated the presence of CD206+ peritoneal dendritic cells in endometrioid lesions, involved in debriding the engraftment area by active phagocytosis [46].

Using a mouse model of endometriosis, Ono Y. et al. (2021) demonstrated the predominance of CD206+ macrophages within the lesions, associated with the activation of angiogenesis (according to VEGFA and TGFβ1 mRNA levels) and growth of the foci [47]. Infiltration of endometrioid lesions by CD206+ macrophages in mouse models has been demonstrated [15,17]; of note, an M1 to M2 shift in macrophage polarization occurred on day 7 after inoculation with endometrial tissue [15]. The hypothesis of a transition from the classical pro-inflammatory macrophage phenotypes (M1) to the alternative anti-inflammatory profile (M2) accompanied by tissue remodeling in endometriosis has been substantiated in animal models. Our data indicate significantly higher systemic representation of CD206+ monocytes in patients with endometriosis vs. the comparison group, signifying their involvement in the pathogenesis.

In our study, adolescent patients with endometriosis revealed lower counts of CD16+ monocytes and lower counts of B1 lymphocytes (CD19^+^CD5^+^) vs. the comparison group. Functionally, CD16^+^ monocytes act as cells patrolling the vascular endothelium, as they eliminate oxidized lipids, immune complexes and dead cells from the endothelium surface. CD16 (FcγRIIIa) is a low-affinity transmembrane receptor that mediates antibody-dependent cellular cytotoxicity, containing two extracellular Ig-like domains. CD16^+^ monocytes express HLA-DR and costimulatory molecules at maximum density, while producing high amounts of pro-inflammatory cytokines TNF-α and IL-12 and decreased amounts of IL-1 and IL-10. Being positive for TLR4, the cells can recognize damage- and pathogen-associated molecular patterns and respond by cytokine synthesis. CD16^+^ monocytes are also capable of antigen presentation [48]. Decreased CD16^+^ monocyte counts in patients with PE may indicate decreased sensitivity to pro-inflammatory patterns, a reduced number of monocytes capable of peripheral tissue colonization and ultimately a decreased anti-lesion protective capacity of the immune system already at the initial stages of the disease. Indeed, we identified the counts of CD16^+^ monocytes (*p* < 0.001) and CD86^+^ monocytes (*p* = 0.017) as significant, independent risk factors for the development of PE in adolescence.

Decreased counts of CD16^+^ NK cells expressing killer cell immunoglobulin-like receptors along with increased counts of highly cytotoxic CD57^+^ NK cells in peritoneal fluid are characteristic of adult patients with endometriosis [49]. Yang H. et al. (2017) recorded a sharp decrease in the expression of CD16, perforin and IFN-γ (molecules associated with NK cell cytotoxicity) when co-culturing endometrial stromal cells from heterotopias and macrophages [50]. Decreased counts of CD16^+^ monocytes in our setting are consistent with the data obtained in adult clinical settings as well as cell culture studies for endometriosis, supporting the impaired NK cell cytotoxicity hypothesis and indicating immune response to endometrioid heterotopias starting from initial stages of the disease.

Indeed, experimental studies in cell culture and animal models, as well as clinical studies in adult patients, identified a mildly dysfunctional state of immunity involving dysregulated interactions of multiple cell entities including neutrophils, macrophages, dendritic cells, natural killer cells, T helper cells and B cells, associated with disease progression [39,50,51]. Our results are overall consistent with the findings obtained in adult patients: adolescents with endometriosis also reveal a decrease in total counts of B1 lymphocytes (CD19^+^CD5^+^). Moreover, B1 cell counts (CD19^+^CD5^+^) were identified as a risk factor for severity of the disease according to rASRM classification. At the same time, T helper cell counts (CD3^+^CD4^+^) were also significantly lower in patients with stages I–II of the disease vs. the comparison group. The declined CD4^+^ T helper functionalities in patients with PE starting from early stages, linked to the overall structure of immune response and stimulation of other immune subsets including cytotoxic T cells and B cells, can be associated with decreased B1 cell counts for the main group. The lower expression of CD4 may compromise the interaction of specific endometrial antigens with major histocompatibility complex class II on antigen-presenting cells and thereby alleviate local responses to the engraftment of endometrial fragments and their integration to ectopic niches in the peritoneum. Furthermore, increased activity of regulatory T (Treg) cells in patients with endometriosis may interfere with the efficiency of recruited immune cells in recognizing and targeting endometrial antigens, thereby supporting the survival of endometrial grafts in the peritoneal cavity and their engraftment after retrograde menstruation [52]. The activity of other immune subsets (macrophages, dendritic cells, NK, CD4^+^ and CD8^+^ lymphocytes) can be suppressed by inducible Treg cells upon chronization of the process [39,53].

Other parameters were statistically similar between the groups, notably the pro-inflammatory marker CD192, a key mediator of monocyte migration and receptor for the monocyte chemoattractant protein CCL2 involved in monocyte infiltration in rheumatoid arthritis, multiple sclerosis and anti-tumor inflammatory response [54]. Still, in patients with endometriosis, the percentage of CD192^+^ cells was higher and significantly correlated with the counts of pro-inflammatory activated CD80^+^ monocytes in the blood (r = 0.805, *p* < 0.001) and in the peritoneal fluid (r = 0.844, *p* < 0.001). In addition, CD192^+^ macrophage counts in the peritoneal fluid positively correlated with serum levels of Hif-1α (r = 0.484, *p* = 0.036) and bound iron (r = 0.496, *p* = 0.031), which supports the hypothesis of monocyte-mediated response to hypoxia and oxidative stress in the peritoneum, characteristic of PE.

The percentage of monocytes positive for other pro-inflammatory markers, HLA-DR and CD80, in blood and peritoneal fluid was also statistically similar between the groups. It is important to note that the HLA-DR major histocompatibility complex class II antigen molecules are expressed by a variety of antigen-presenting cells ranging from macrophages and dendritic cells to B lymphocyte subsets capable of antigen presentation [55]. The participation of HLA-DR^+^ monocytes in pro-inflammatory activation was indirectly confirmed by correlating the counts of HLA-DR^+^ monocytes and CD45^+^, CD80^+^ and CD86^+^ monocyte subsets observed identified by us in this study.

CD80 is a type 1 transmembrane glycoprotein constitutively expressed on B and T cells, as well as macrophages and monocytes, playing an important role in antitumor signaling. We identified a trend toward an increased percentage of CD80^+^ cells in PE, apparently linked to the inflammatory component of the disease, albeit below the threshold of significance [56,57].

Elevated levels of common inflammatory markers in adult patients with endometriosis, including WBC count, neutrophil-to-lymphocyte ratio, platelet count, mean platelet volume and platelet distribution width, as well as platelet-to-lymphocyte ratio and CA125 level, were reported previously, particularly from a diagnostic perspective [58]; however, we observed no significant differences in these parameters between the PE and comparison groups of this study.

The comparative characterization of macrophage subsets performed by us in this study involved the conventional distinction between M1 macrophages (the classically activated pro-inflammatory cells, CD80^+^/CD86^+^/HLA-DR^+^/CD192^+^) and M2 macrophages (the alternatively activated pro-regenerative anti-inflammatory CD163^+^/CD206^+^ subset). The tendency toward the M1 polarization of macrophages in endometriosis has been associated with the increased production of netrin-1, a neuronal guidance signaling molecule known to also support neoangiogenesis by enhancing endothelial cell motility and morphogenetic potential [59,60]. Dysregulation of monocyte/macrophage pools characteristic of endometriosis [61] may promote a decline in the rates of T cell activation and cytotoxicity, starting from the initial stages of the disease in adolescents.

After treatment, adolescent patients with PE had an increase in CD206^+^ monocyte counts and a decrease in CD163^+^ cell counts. CD163 is a scavenger receptor; indeed, the need for the utilization of hemolysis products and hemoglobin in patients against the background of developing therapeutic amenorrhea was lower than before treatment, especially considering the incidence of heavy menstruation and retrograde menstrual flow episodes associated with physiological uterine flexion characteristic of childhood. The observed increase in CD206+ monocyte counts during treatment (vs. the comparison group) may be indirectly related to the ongoing repair processes accompanied by the formation of an immunomodulatory background. Also indirectly, a decrease in the level of CD206^+^ monocytes in the blood of patients with treatment-related amenorrhea may have indicated the prospective utility of this indicator as a marker of clinical efficacy. The several-fold increased CD206^+^ monocyte counts in patients with endometriosis after the therapy as compared with the control group suggest insufficiency of the one-year course for long-term pain relief and prevention of progression in PE. Zutautas K. et al. (2023) demonstrated a decrease in CD206^+^ macrophage counts during treatment for PE in a mouse model [62] and associated this trend with suppressed vascularization of the foci. Based on dynamics of anti-inflammatory macrophage subsets of the blood before and after surgical treatment for PE, Sun S. et al. (2022) hypothesized that M2 polarization of macrophages plays significant role in the recurrence of endometrioid cysts one year after surgery [63].

In this study, we also assessed lymphocyte subset dynamics before and after the therapy, and with regard to the comparison group. Natural killer (NK) cells were of particular interest in connection with their ability to eliminate endometrioid cells. NK cells expressing surface marker CD56 at a high and low density (respectively, CD56^bright^ and CD56^dim^) have different functional properties: CD56^bright^CD16^–/dim^ cells are considered efficient producers of cytokines with immunoregulatory properties but can become cytotoxic when activated, whereas CD56^dim^ NK cells exert higher cytotoxicity by expressing more immunoglobulin-like receptors as well as Fcγ receptors CD16. In adult patients with endometriosis, NK cell activity and cytotoxicity toward autologous endometrial cells is reduced and correlates with disease severity [39].

Before the start of the therapy, the content of NK cells did not differ significantly between the study groups. After the therapy, patients of the main group showed a sharp increase in the percentage of cytotoxic CD56dimCD16bright NK cells in comparison with the data before the therapy. Factor analysis for pain relief efficacy in adolescents during one-year treatment for PE revealed the significant influence of the cytotoxic CD56dimCD16bright cell counts both before (*p* < 0.001) and after the treatment (*p* = 0.006). Prediction accuracy for the clearance of pain symptoms in patients with PE after 12 months on dienogest using the percentage of cytotoxic CD56dimCD16bright NK cells before therapy was 91.3% (Chi^2^ = 11.67, *p* < 0.001), indicating the prospective clinical utility of this parameter as a predictor of treatment efficacy and life quality in young patients.

It is noteworthy that the counts of both cytotoxic CD56dimCD16bright (*p* = 0.006) and immunoregulatory CD56brightCD16dim (*p* = 0.007) NK cells, and their total (*p* = 0.004), were higher in patients with pain symptoms persisting during the therapy, likely reflecting a resilient pathogenic process after 12 months of therapy. The persistently higher counts of CD56brightCD16dim NK cells vs. the comparison group (0.007 ± 0.003 and 0.004 ± 0.004, *p* = 0.077) highlights the need for longer-term hormonal therapy for PE in adolescents.

The decreased cytotoxic activity of NK cells in adult patients with PE as a putative factor interfering with the targeted destruction of lesions by the immune system has been elucidated previously [50,53,64]. However, the possibility of using NK levels as a marker of therapeutic efficacy had not been considered, despite the consideration of the prospective use of NK cells for immunotherapy in PE [65]. Increased counts of CD56^+^ lymphocytes after the treatment reflect the activation of the immune response and its NK-dependent nature, while the positive correlation between CD56brightCD16dim NK cell counts and the diagnosed stage of the disease (r = 0.498, *p* = 0.022) or rASRM score (r = 0.520, *p* = 0.016) indicate activation of cell-mediated immunity during the treatment, notably in patients with an aggravated course of the pathological process.

Adolescent patients with PE (main group) showed a therapy-related decrease in the neutrophil-to-lymphocyte ratio (NLR, *p* = 0.008). At the same time, the NLR for the main group was statistically similar to that for the comparison group before and after the therapy, and no differences among subgroups based on severity of the disease were recorded. After the therapy, patients of the main group showed a decrease in the neutrophil-to-lymphocyte ratio (NLR, *p* = 0.008). At the same time, the NLR for the main group was statistically similar to that of the comparison group before and after the therapy, and no differences among subgroups based on the severity of the disease were recorded. In a meta-analysis by Tabatabaei F. et al. (2023) the NLR was identified as a prospective diagnostic marker for PE with fairly high sensitivity and specificity in adult patients (respectively, 67% and 68%), but also lacking significant associations with severity [66].

Peripheral serum levels of TGF-β and soluble markers for hypoxia (HIF-1α) and angiogenesis (VEGF-A, sVEGFR2), significantly increased in adult forms of endometriosis [37,48,50], were similar in adolescents with and without PE (the main vs. comparison groups of this study). At the same time, we identified correlations between the serum levels of TGF-β1 and overall counts of CD3^–^CD56^+^16^+^ NK cells in the blood, as well as between the serum levels of TGF-β3 and CD55^bright^CD16^–/dim^NK cell counts. TGF-β promotes differentiation and the induction of Treg cells, and low Treg counts in peritoneal fluid of patients indicate the disruption of these TGF-β dependent processes in PE [49]. TGF-β secreted by macrophages can reduce cytotoxicity and inhibit the metabolic activity of NK cells, thereby supporting a switch to aerobic glycolysis and acidification of the ectopic foci [39,50].

The main findings of this study are summarized in Figure 5.

Thus, our findings confirm the hypothesis of altered blood mononuclear cell composition in endometriosis, starting from the onset of the disease in adolescents. The observed imbalance in the contents of pro- and anti-inflammatory activated monocytes suggests the degree of immunological shifts as a risk factor for the disease severity. One-year conservative treatment with progestogens promoted an increase in NK cell counts against the background of continued high anti-inflammatory activity of monocytes, correlating with pain relief and treatment-related amenorrhea indicators of efficacy.

## 5. Conclusions

The blood counts of several subsets of mononuclear leukocytes are specifically altered in peritoneal endometriosis. The alterations, which include increased counts of monocyte subsets positive for pro-inflammatory (CD86) and anti-inflammatory markers (CD206) and lower counts of CD16^+^ monocytes and B1 lymphocytes (CD19^+^CD5^+^) indicate that characteristic immunological profiles are already present at initial stages of the disease in adolescent patients.In the blood of adolescents with endometriosis, the level of CD86+ monocytes is higher, and CD16^+^ is lower under the influence of the disease itself (F = 6.84, *p* = 0.003) under the condition of persistent dysmenorrhea (F = 15.36, *p* < 0.001) and chronic pelvic pain (F = 14.36, *p* < 0.001). In the peritoneal fluid, under the influence of the disease, the level of CD86^+^ is higher and the level of CD16^+^ macrophages is lower (F = 4.74, *p* = 0.033). At the same time, blood levels of CD16^+^ monocytes (*p* < 0.001, OR 35.00, percent correct 85.00%) and CD86^+^ monocytes (*p* = 0.017, OR 5.33; percent correct 70.00%) are significant independent risk factors for developing the disease in adolescence.In adolescents with endometriosis receiving progestogens for 1 year, a higher proportion of cytotoxic lymphocytes with the CD56dimCD16bright phenotype (*p* = 0.049), a higher level of CD206^+^ monocytes (*p* < 0.001) and a lower level of anti-inflammatory CD163^+^ monocytes (*p* = 0.017) was observed, which likely mediated the clinical effect of reducing the number of endometrioid heterotopias during therapy. However, the content of CD206^+^ and CD163^+^ monocytes during treatment after 12 months did not normalize and differed significantly from the content in the comparison group, which, along with the clinical picture, can serve as a justification for longer-term hormonal therapy for endometriosis in adolescents.A higher percentage of cytotoxic CD56dimCD16bright cells in the blood of patients with endometriosis before treatment (OR 5.25, DI 0.44; 62.12, *p* < 0.001) and during treatment (OR 1.92, DI 0.94; 3.93, *p* = 0.006), as well as the content of CD206+ monocytes during treatment (OR 0.82, DI 0.62; 1.08, *p* = 0.038), determined the effectiveness of relieving pain symptoms in adolescence after 1 year of treatment of endometriosis with progestogens.

## Figures and Tables

**Figure 1 cells-13-01187-f001:**
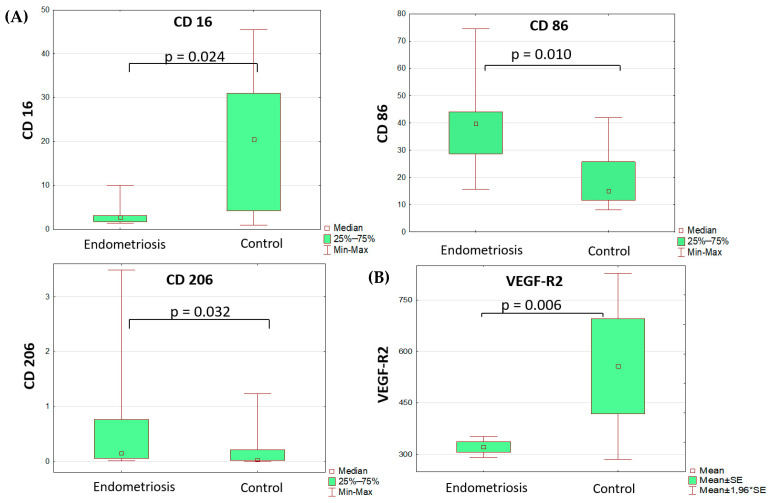
Differences in monocyte subset composition (**A**) of the blood (Mann–Whitney test) and sVEGFR2 (**B**) serum levels (*t*-test) in adolescents with PE and the comparison group.

**Figure 2 cells-13-01187-f002:**
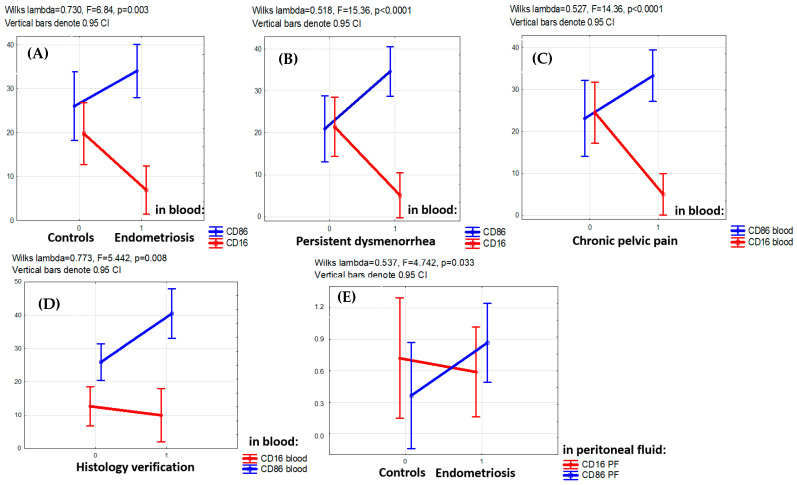
Factor analysis for CD16+ and CD86+ monocyte blood counts with regard to clinical indicators. (**A**,**B**) Multidirectional factorial influence of the presence of the disease itself (**A**), persistent dysmenorrhea (**B**), chronic pelvic pain (**C**) in adolescents on the levels of CD16+ (decreased level) and CD86+ (increased level) monocytes in the blood (factorial ANOVA). (**D**) Lower level of CD16+ and higher level of CD86+monocytes in the blood in cases when the disease is confirmed by histological analysis. (**E**) Lower levels of CD16+ and higher CD86+macrophages in the peritoneal fluid (PF) in patients with endometriosis versus the comparison group (factorial ANOVA).

**Figure 3 cells-13-01187-f003:**
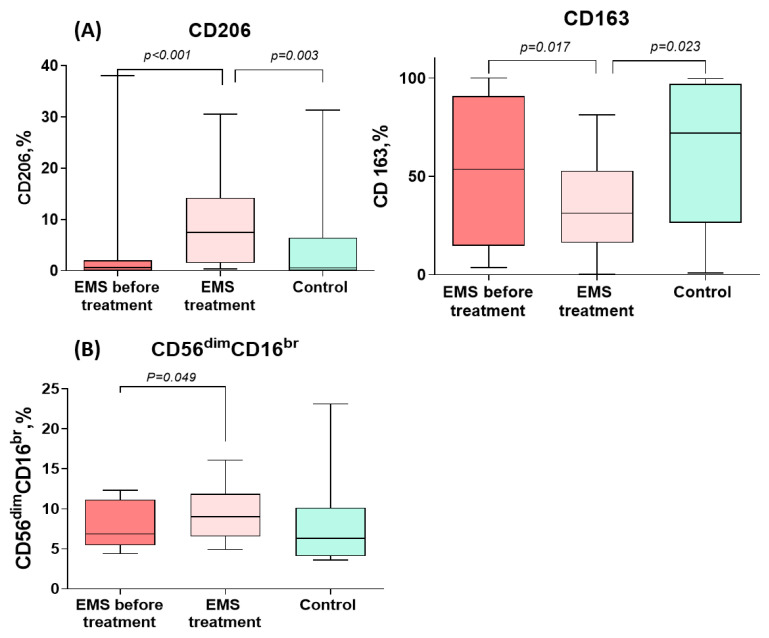
Comparative dynamics of monocyte (**A**) and lymphocyte (**B**) subsets in patients with peritoneal endometriosis (PE) on dienogest. The data are presented as the Me (27–75%), min–max; *p*-values calculated for PE before vs. after treatment, Wilcoxon test; PE before treatment vs. comparison group, Mann–Whitney test.

**Figure 4 cells-13-01187-f004:**
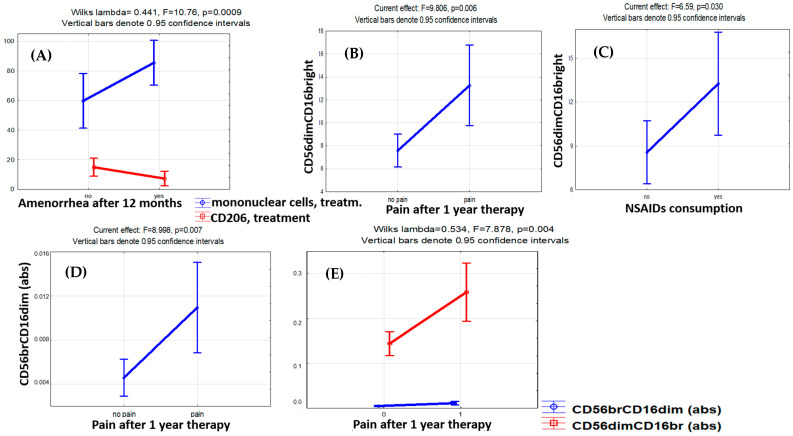
Factor analysis for monocyte and lymphocyte subset counts with regard to therapy efficacy indicators. The influence of therapy effectiveness factors on the levels of monocytes and lymphocytes. (**A**) Increase in mononuclear cells (CD45+) and decrease in CD206+ monocytes in patients with amenorrhea after 12 months of therapy. (**B**) Higher level of cytotoxic CD56dimCD16bright NK cells with persistent pain and (**C**) the need to take NSAIDs to relieve pain in patients after 12 months of therapy. (**D**) In the case of pain persistence in patients with endometriosis after 12 months of therapy, the level of CD56brightCD16dim NK cells with immunoregulatory properties in the blood appeared to be higher. (**E**) Unidirectional dynamics in blood levels of CD56brightCD16dim and CD56dimCD16bright NK cells in the case of pain persistence after 12 months of therapy.

**Figure 5 cells-13-01187-f005:**
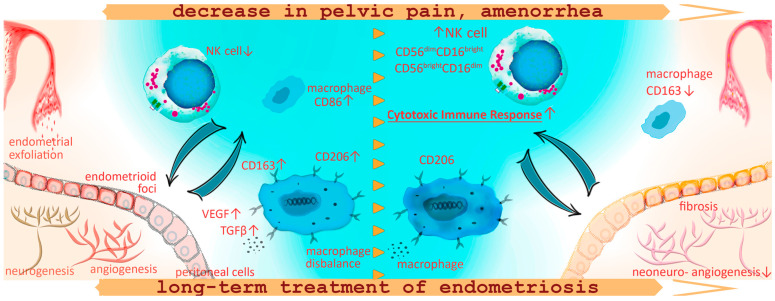
Schematic representation of the macrophage and lymphocyte phenotype alterations against the background of pain reduction and amenorrhea development during the treatment for peritoneal endometriosis with progestogens in adolescents. In adolescent patients with PE, the decreased CD16+ monocyte counts may indicate decreased sensitivity to pro-inflammatory patterns, a reduced number of monocytes capable of peripheral tissue colonization, clearance of cell debris from menses and ultimately a decreased anti-lesion protective capacity of the immune system already at the initial stages of the disease. Increased counts of CD86+ cells in patients with endometriosis, as well as the association of these pro-inflammatory monocytes with chronic pelvic pain and persistent dysmenorrhea, indicate extensive inflammatory activation and systemic immune response in PE. Immune imbalance is associated with the maintenance of the endometriotic lesion by promoting neoangio- and neurogenesis and fibrotic adhesions. One-year conservative treatment with progestogens according to our findings promoted an increase in NK cell counts in peripheral blood reflecting probable activation of the peritoneal cytotoxic immune response for endometrial fragment clearance in the peritoneal cavity, decreasing the potential implantation of ectopic implants and endometriosis regression. At the same time, a year of hormonal therapy was accompanied with continued high anti-inflammatory activity of monocytes, CD+206 monocytes to be exact, responsible for vascularization and endometrioid lesion progression. Monocyte counts positive for scavenger receptor CD163 decreased during the treatment, possibly in connection with treatment-related amenorrhea and the absence of retrograde menstruation, indirectly indicating a decrease in oxidative stress levels within the peritoneal cavity.

**Table 1 cells-13-01187-t001:** Mononuclear cell and soluble factor profiles at the time of laparoscopy.

Parameter	Main Group (PE), *n* = 50	Comparison Group, *n* = 20	*p*-Value
Monocyte subsets, blood			
Monocytes, abs, 10^9^ cells/L	0.57 (0.42–0.76)	0.64 (0.52–0.91)	0.178
Monocytes, %	9.25 (7.80–10.70)	9.05 (6.40–9.45)	0.893
CD192+, %	28.01 (8.92–40.88)	14.89 (6.39–20.96)	0.237
HLA-DR+, %	50.16 (6.99–92.30)	57.84 (47.78–97.10)	0.470
CD80+, %	3.31 (1.6–8.51)	0.92 (0.44–4.65)	0.260
CD206+, %	0.15 (0.05–0.76)	0.03 (0.02–0.22)	0.032
CD163+, %	80.56 (14.90–96.62)	90.10 (53.60–98.00)	0.266
CD56+, %	0.85 (0.50–1.20)	1.00 (0.40–1.85)	0.789
CD16+, %	2.90 (1.80–3.20)	18.90 (2.45–35.00)	0.024
CD86+, %	39.75 (28.70–44.10)	15.00 (11.60–25.80)	0.010
Monocyte subsets, peritoneal fluid			
CD192+, %	8.62 (6.25–10.66)	12.81 (6.99–14.52)	0.523
HLA-DR+, %	90.99 (78.57–98.53)	84.29 (69.31–97.87)	0.898
CD206+, %	0.36 (0.21–0.72)	0.53 (0.20–0.59)	0.766
CD80+, %	0.47 (0.16–1.44)	1.01 (0.24–2.00)	0.594
CD163+, %	95.94 (75.77–98.55)	96.40 (80.24–98.05)	0.879
CD56+, %	0.64 (0.20–0.90)	1.05 (0.45–1.45)	0.522
CD16+, %	0.60 (0.15–0.90)	0.50 (0.25–1.25)	0.831
CD86+, %	0.80 (0.40–1.10)	0.30 (0.16–0.65)	0.201
Lymphocyte subsets, blood			
Leukocytes, abs, 10^9^ cells/L	5.14 (4.28–6.51)	5.54 (4.99–6.18)	0.329
Lymphocytes, %	36.50 (29.10–43.20)	35.60 (32.35–39.55)	0.956
Lymphocytes, abs	1.81 (1.48–2.13)	1.88 (1.85–2.15)	0.143
CD3+, %	73.20 (70.10–78.30)	73.30 (70.30–75.25)	0.947
CD3+, abs	1.33 (1.11–1.56)	1.39 (1.33–1.53)	0.226
CD3+CD4+	41.60 (37.30–46.45)	43.95 (39.0–46.8)	0.374
CD3+CD4+, abs	0.74 (0.56–0.89)	0.87 (0.76–0.93)	0.055
CD3+CD8+, %	26.35 (23.70–31.45)	26.20 (23.25–28.30)	0.579
CD3+CD8+, abs	0.49 (0.39–0.64)	0.50 (0.46–0.56)	0.602
CD3+CD56+CD16+, %	1.10 (0.70–1.60)	1.05 (0.55–1.30)	0.520
CD3+CD56+CD16+, abs	0.02 (0.01–0.03)	0.02 (0.01–0.03)	0.868
CD3-CD56+CD16+, %	8.10 (5.60–10.50)	5.75 (3.7–9.8)	0.464
CD3-CD56+CD16+, abs	0.14 (0.09–0.21)	0.12 (0.07–0.23)	0.816
CD56brightCD16dim, %	0.30 (0.20–0.55)	0.15 (0.10–0.40)	0.140
CD56brightCD16dim, abs	0.01 (0.0–0.01)	0.004 (0.002–0.01)	0.179
CD56dimCD16bright, %	7.55 (6.4–11.65)	7.1 (4.50–9.15)	0.617
CD56dimCD16bright, abs	0.15 (0.1–0.22)	0.15 (0.09–0.19)	0.938
CD19+, %	12.50 (10.80–15.10)	12.60 (11.90–15.70)	0.418
CD19+, abs	0.23 (0.16–0.30)	0.26 (0.21–0.32)	0.202
CD19+CD5+, %	1.10 (0.65–2.10)	1.60 (1.35–1.70)	0.095
CD19+CD5+, abs	0.02 (0.01–0.04)	0.03 (0.02–0.04)	0.049
PAN, %	98.80 (97.80–99.10)	99.15 (98.45–99.35)	0.081
SI	45.00 (32.00–53.00)	47.50 (38.00–56.50)	0.484
Neutrophils, %	53.65 (46.56–60.35)	49.05 (47.65–56.55)	0.594
Neutrophils, abs	2.69 (2.14–3.45)	2.74 (2.23–3.60)	0.665
NLR	1.44 (1.06–1.95)	1.36 (1.24–1.75)	0.851
Soluble factors, blood			
HIF-1α, pg/mL	13.49 (0.81–61.17)	11.63 (0.92–56.89)	0.482
TGF-β1, pg/mL	28.30 (7.02–92.77)	8.50 (6.38–65.82)	0.220
TGF-β2, pg/mL	43.95 (4.65–98.80)	36.46 (3.81–86.16)	0.441
TGF-β3, pg/mL	49.55 (30.84–75.03)	48.62 (20.51–65.73)	0.336
VEGF-A, pg/mL *	23.61 ± 2.25	24.99 ± 3.27	0.028
sVEGFR2, pg/mL *	322.47 ± 126.01	556.50 ± 547.78	0.001

Note: for non-normally distributed variables, data are presented as the Me (Q1–Q3), Mann–Whitney test; for * normally distributed variables, data are presented as the M ± SD, *t*-test. Abbreviations: PE, peritoneal endometriosis; PAN, phagocytic activity of neutrophils; SI, stimulation index; NLR, neutrophil-to-lymphocyte ratio; VEGF-A, vascular endothelial growth factor A; sVEGFR2, soluble VEGF receptor 2; HIF-1α, hypoxia-induced factor 1α; TGF-β, transforming growth factor β, TGF-β1, TGF-β2 and TGF-β3 isoforms.

**Table 2 cells-13-01187-t002:** Clinical and laboratory characterization of adolescent patients with peritoneal endometriosis at diagnosis vs. 12 months on dienogest since the diagnosis.

Parameter	PE, Before Treatment (*n* = 24)	PE, 1 Year on Progestogens (*n* = 24)	Reference Values	*p*-Value
VAS, score	8.00 (7.00–10.00)	0.00 (0.00–3.00)	<4	<0.001
Hemoglobin, g/L	126.0 (121.0–133.0)	134.2 (129.5–139.0)	117–145	<0.001
Erythrocytes, 10^12^ per L	4.54 (4.38–4.73)	4.68 (3.93–5.23)	3.8–4.7	0.025
Hematocrit, L/L	0.38 (0.37–0.39)	0.40 (0.37–0.41)	0.34–0.45	0.086
Platelets, 10^9^ per L	268.0 (227.0–307.0)	236.0 (226.0–270.0)	150–400	0.725
Mean platelet volume, fl	9.95 (9.40–10.40)	10.0 (9.7–10.5)	9.4–12.3	0.763
Total iron (Fe2+), µmol/L	20.50 (13.80–27.60)	19.20 (17.32–21.43)	8.8–27	0.779
Ferritin, µg/L	21.30 (12.50–36.40)	35.35 (26.20–50.80)	20–250	0.007
Fibrinogen, g/L	3.57 (2.37–3.10)	2.61 (2.31–2.88)	1.7–3.7	0.965
CRP, mg/L	0.62 (0.32–1.44)	0.43 (0.24–0.90)	0–5	0.259
LH, IU/L	5.85 (4.00–8.40)	5.75 (3.40–7.90)	2.4–5.4	0.156
FSH, IU/L	5.50 (4.30–7.00)	6.30 (5.07–7.50)	1.9–3.7	0.492
Prolactin, mlU/L	362.00 (250.00–549.00)	368.00 (289.00; 492.00)	226–502	0.566
Estradiol, pmol/L	200.35 (132.90–407.40)	104.03 (55.74–245.00)	188–335	0.028
Testosterone, nmol/L	0.92 (0.74–1.46)	0.80 (0.64–1.03)	1.2–1.9	0.374
FAI, %	1.72 (1.04–2.35)	1.83 (1.04–2.70)	<4.5	0.676
Cortisol, nmol/L	416.50 (332.50–517.00)	345.00 (257.00–401.00)	212–469	0.008
17-OHP, nmol/L	5.10 (3.00–8.10)	3.30 (2.50–3.90)	1.24–7.11	0.020
DHEAS, µmol/L	4.59 (3.15–5.90)	4.61 (3.48–5.86)	0.9–11.7	0.064
Androstenedione, ng/mL	10.40 (6.67–12.75)	6.79 (3.85–7.81)	1–12.2	0.026
AMH, ng/mL	3.57 (2.64–4.88)	3.23 (2.52–4.53)	0–10.6	0.223
CA-125, U/mL	21.21 (14.26–34.66)	10.31 (7.17–14.90)	0–35	<0.001
CA-19.9, U/mL	6.29 (3.96–11.94)	6.78 (4.21–10.76)	0–37	0.051
HE4, pmol/L	48.33 (42.85–54.23)	47.41 (41.34–51.96)	0–60	0.115

Note: For variables with a non-normal distribution, data are presented as the median (25–75 percentiles), Wilcoxon matched pairs test. LH: luteinizing hormone; FSH: follicle-stimulating hormone; CRP: highly sensitive C-reactive protein, HE4: human epididymis protein 4; CA-125: carbohydrate antigen 125; 17OHP: 17OH-progesterone; FAI: free androgen index; AMH: anti-Mullerian hormone.

**Table 3 cells-13-01187-t003:** One-year progestogen therapy-related dynamics of mononuclear cell and soluble factor profiles in adolescents with peritoneal endometriosis (PE).

Parameter	PE, before Treatment	PE, 1 Year on Progestogens	*p*-Value
Monocyte subsets (*n* = 24)			
Monocytes, 10^9^ cells/L	0.57 (0.42–0.76)	0.46 (0.37–0.54)	0.316
Monocytes, %	9.25 (5.60–14.60)	8.90 (8.50–10.25)	0.987
CD206+, %	0.76 (0.09–2.70)	7.50 (1.55–14.20)	<0.001
CD163+, %	45.02 (16.27–90.35)	31.20 (16.30–52.80	0.017
CD16+, %	2.90 (1.84–6.30)	6.16 (2.24–22.90)	0.686
CD86+, %	32.65 (24.83–44.00)	46.30 (32.10–70.80)	0.345
Lymphocyte subsets (*n* = 11)			
Leukocytes, 10^9^ cells/L	5.99 (4.71–6.53)	5.31 (4.87–6.09)	0.891
Lymphocytes, %	36.90 (29.40–43.10)	35.70 (30.30–36.60)	0.401
Lymphocytes, abs	1.78 (1.46–2.84)	1.97 (1.63–2.18)	0.631
CD3+, %	72.70 (70.40–75.60)	73.30 (71.10–76.30)	0.678
CD3+, abs	1.40 (1.22–1.75)	1.35 (1.24–1.60)	0.634
CD3+CD4+	41.10 (38.80–47.20)	43.00 (35.80–45.50)	0.101
CD3+CD4+, abs	0.84 (0.71–1.90)	0.75 (0.68–0.87)	0.523
CD3+CD8+, %	26.40 (21.80–30.10)	23.30 (25.30–31.80)	0.597
CD3+CD8+, abs	0.46 (0.39–0.52)	0.45 (0.41–0.63)	0.410
CD3+CD56+CD16+, %	1.20 (0.70–1.50)	1.40 (0.95–1.63)	0.321
CD3+CD56+CD16+, abs	0.02 (0.02–0.03)	0.02 (0.02–0.03)	0.386
CD3+CD56+CD16+, %	8.00 (7.70–8.60)	7.30 (6.10–9.80)	0.112
CD3^−^CD56^+^CD16^+^, abs	0.14 (0.12–0.21)	0.16 (0.11–0.17)	0.611
CD56brightCD16dim, %	0.40 (0.20–0.60)	0.30 (0.20–0.50)	0.789
CD56brightCD16dim, abs	0.01 (0.00–0.01)	0.01 (0.0–0.10)	0.091
CD56dim CD16bright, %	6.90 (5.91–10.08)	9.20 (6.40–14.00)	0.049
CD56dim CD16bright, abs	0.17 (0.11–0.22)	0.19 (0.14–0.25)	0.382
CD19+, %	12.70 (12.50–13.40)	12.40 (9.40–14.50)	0.210
CD19+, abs	0.22 (0.19–0.36)	0.24 (0.21–0.27)	0.312
CD19+CD5+, %	1.20 (0.70–2.10)	1.30 (0.90–1.80)	0.785
CD19+CD5+, abs	0.03 (0.02–0.05)	0.03 (0.01–0.05)	0.864
PAN, %	98.80 (98.40–99.10)	98.80 (97.80–99.20)	0.201
SI	53.00 (39.00–54.00)	43.00 (33.00–54.00)	0.832
Neutrophils, %	48.50 (45.50–59.80)	55.10 (49.80–55.90)	0.324
Neutrophils, 10^9^ cells/L	2.93 (2.37–3.91)	2.56 (2.39–3.32)	0.678
NLR	1.44 (1.06–1.89)	0.08 (0.06–0.10)	0.008

Note: for variables with a non-normal distribution, data are presented as the median (25–75 percentiles), Wilcoxon matched pairs test. PAN—phagocytic activity of neutrophils; SI—stimulation index; NLR—neutrophil-to-lymphocyte ratio.

## Data Availability

The materials and results of this research contain personal data of patients and are available on request from the corresponding author.

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
