# Peer review of "Altered Monocyte and Lymphocyte Phenotypes Associated with Pathogenesis and Clinical Efficacy of Progestogen Therapy for Peritoneal Endometriosis in Adolescents"

_cells, 2024, doi:10.3390/cells13141187_

Round 1

Reviewer 1 Report

Comments and Suggestions for Authors

The study aimed to assess changes in immune cell populations in adolescents with peritoneal endometriosis pre- and post-1-year progestogen therapy, highlighting specific monocyte and lymphocyte subsets relevant to the disease and treatment response, endorsing prolonged hormonal therapy. The reviewer suggests using monocyte subsets and soluble factors as potential biomarkers, noting a significant effect (10x) on CD206 post-1-year progestogen therapy. The manuscript, particularly the discussion, should be condensed and simplified, with structured subtitles. Avoid repeating results in the discussion, contextualize observations, and generalize the take-home message in the conclusions.

MAJOR COMMENTS (CRITICAL)

1. Clarify the discrepancy in the number of patients in the control group: on page 2, "paramesonephric cysts (n=20, the comparison group)" versus "The comparison group consisted of 15 adolescent girls." Justify the choice of 50 versus 20 (or 15): was it statistically determined for analytical power?

2. Notably, only a subgroup of patients (n=24) received 12 months of dienogest. The selection criteria for these patients are not specified in the methods, potentially introducing bias. Was randomization employed? The analysis is paired pre- and post-therapy. Why wasn't a comparison made between patients treated for 1 year and those untreated for 1 year? Clarifications are needed here.

SPECIFIC COMMENTS

1. Incorporate CD206, CD163 (and CD86) into the introduction rather than the discussion.

2. Clearly state the study's aim in the body of text (as done in the abstract). Include a hypothesis and rationale, articulating, "In this study, we address... ".

3. Include a reference to the revised American Society for Reproductive Medicine (rASRM) criteria, specifying the date of revision.

4. Specify the temperature and time for fixing cell pellets with 2% paraformaldehyde. Explain the choice of 2% PAF for blood isolation (page 3) and 4% PAF for peritoneal fluid isolation (page 4).

5. Address the issue of testing numerous parameters and computing p-values for each. Were the p-values corrected for multiple testing (q-value)?

6. Elaborate on the inclusion of parameters carbohydrate antigen 125 (CA-125) and CA-19.9, explaining their significance in endometriosis management.

7. Refrain from presenting non-significant data as "lower" or indicating trends without statistical significance to prevent potential misinterpretation of data.

8. Consider adapting Figure 1 into violin plots, especially for CD206, and enhance Figure 3.

9. Improve the presentation of Figure 4, look for consistency.

10. In Table 2, provide reference values in the left column for clarity.

11. Provide a reference for the statement "NK cells known to produce cytokines with immunoregulatory properties."

12. Specify which metalloproteinases (MMP).

13. Add references for the paragraphs on CD86 (page 14) and CD80 (page 16).

Comments on the Quality of English Language

PRESENTATION AND LANGUAGE

Consistently present p-values as p=0.xxx in the abstract and text. Specify which markers indicate M1/M2 polarization.

Improve English grammar, correct typos, and translate "Железо" to English.

Maintain consistency in using exponent notation (e.g., CD56brightCD16-/dim) throughout the manuscript for readability.

Include units for general blood test parameters, biochemical, and hormonal profiles.

Use the official name FlowJo, LLC when referencing.

Enhance referencing format, e.g., ([7–10], [11], [5–12]) or ([13][12–14–15]).

Always provide line and page numbers for peer-reviewing documents.

Author Response

  1. Summary

Dear reviewer,

First of all, let me, on behalf of the team of authors and myself, thank you for your deep and constructive analysis of the manuscript we submitted, which allowed us to improve it.  We appreciate the constructive criticism and will answer all questions step by step.

Indeed, according to our results, the characteristics of immunoprofiling of lymphocytes (cytotoxic NK-cells) and monocytes (CD206+) can be considered as potential biomarkers of the clinical effectiveness of progestogen therapy for peritoneal endometriosis in adolescents. As well, as blood levels of CD16+ monocytes (p<0.001, OR 35.00, percent correct 85.00%) and CD86+ monocytes (p=0.017, OR 5.33; percent correct 70.00%) are significant independent risk factors for developing the disease in adolescence and could be used as potential diagnostic markers with 70-85% accuracy in adolescents.  

We actually recalculated diagnostic significance for each marker and immunological aspects also, however, the significance was inferior to pain assessment scale using VAS in combination with MRI criteria or indicators of pain chronicity and persistence of dysmenorrhea. We summarized the results of different clinical features with diagnostic values in previous article (Khashchenko, E.P.; Uvarova, E.V.; Fatkhudinov, T.K.; Chuprynin, V.D.; Asaturova, A.V.; Kulabukhova, E.A.; Vysokikh, M.Y.; Allakhverdieva, E.Z.; Alekseeva, M.N.; Adamyan, L.V.; Sukhikh, G.T. Endometriosis in Adolescents: Diagnostics, Clinical and Laparoscopic Features. J. Clin. Med. 2023, 12, 1678. https://doi.org/10.3390/jcm12041678).

As for our results of monocytes and lymphocytes levels, the most perspective results we assessed for the percentage of cytotoxic CD56dimCD16bright NK cells both before treatment (OR 5.25, DI 0.44–62.12, p<0.001) and after treatment (OR 1.92, DI 0.94–3.93, p=0.006) as significant predictors of the pain relief efficacy after a year of treatment (logistic regression analysis) (page 13). Also we observed treatment-related dynamics of CD206+ monocytes in the blood which was associated with pain relief efficacy (OR 0.82, DI 0.62–1.08, Chi2=4.29, p=0.038), playing an essential role in cell metabolic reprogramming and transition to glycolysis as the dominant energy production mode, concomitant acidosis, and angiogenesis in the foci.

We summarized our results in the figure 5.

Lines 681-698 Page 19: In adolescent patients with PE the decreased CD16+monocyte counts may indicate decreased sensitivity to pro-inflammatory patterns, reduced number of monocytes capable of peripheral tissue colonization, clearance of cell debris from menses and ultimately a decreased anti-lesion protective capacity of the immune system already at initial stages of the disease. Increased counts of CD86+cells in patients with endometriosis, as well as association of these proinflammatory monocytes with chronic pelvic pain and persistent dysmenorrhea, indicates extensive inflammatory activation and systemic immune response in PE. Immune imbalance is associated with the maintenance of the endometriotic lesion by promoting neoangio-/neurogenesis and fibrotic adhesions. One-year conservative treatment with progestogens according to our findings promoted an increase in NK cell counts in peripheral blood reflecting probable activation of peritoneal cytotoxic immune response for endometrial fragment clearance in the peritoneal cavity, decreasing the potential implantation of ectopic implants and endometriosis regression. At the same time a year of hormonal therapy was accompanied with continued high anti-inflammatory activity of monocytes, exactly CD+206 monocytes, responsible for vasculariza-tion and endometrioid lesions progression. Monocyte counts positive for scavenger receptor CD163 decreased during the treatment, possibly in connection with treatment-related amenor-rhea and the absence of retrograde menstruation, indirectly indicating a decrease in oxidative stress levels within the peritoneal cavity.

Thank you very much for taking the time to review this manuscript and the profound interest in our study. Please find the detailed responses below and the corresponding revisions in track changes in the re-submitted files.

MAJOR COMMENTS (CRITICAL)

  1. Clarify the discrepancy in the number of patients in the control group: on page 2, "paramesonephric cysts (n=20, the comparison group)" versus "The comparison group consisted of 15 adolescent girls." Justify the choice of 50 versus 20 (or 15): was it statistically determined for analytical power?

Response 1. Thank you for this correction. The number of patients in the comparison group was 20, as it was stated in the abstract section and Methodology.

Lines 89-90: We corrected in the text, thank you: The comparison group consisted of 20 adolescent girls of the same age (16.0 (15.0–17.0)).

We calculated the sample size to obtain statistically correct results was based on literature data on the likelihood of the association of endometriosis with immunological dysfunction and pro-inflammatory status in adults (because there are not similar publications in adolescents), using the STATISTICA 10 program.

For example:

Based on the literature data on the relationship between endometriosis and altered monocytes profiling (article Takebayashi, A., Kimura, F., Kishi, Y., Ishida, M., Takahashi, A., Yamanaka, A., … Murakami, T. (2014). Subpopulations of Macrophages within Eutopic Endometrium of Endometriosis Patients. American Journal of Reproductive Immunology, 73(3), 221–231. doi:10.1111/aji.12331) AND literature data on the cytometry and immunohistochemistry data in the patients with endometriosis (Hey-Cunningham AJ, Wong C, Hsu J, Fromm PD, Clark GJ, Kupresanin F, Miller EJ, Markham R, McGuire HM. Comprehensive analysis utilizing flow cytometry and immunohistochemistry reveals inflammatory changes in local endometrial and systemic dendritic cell populations in endometriosis. Hum Reprod. 2021 Jan 25;36(2):415-428. doi: 10.1093/humrep/deaa318. PMID: 33313846) Assuming an alpha level of 0.05 and a study confidence level of 90% (Independent Sample t-Test), when calculating for each parameter based on M ± SD, it is necessary to include from 5 to 12 patients in each group. Considering that 15% of patients may drop out of the study, the sufficient sample size will be: 14 patients in the group, a total of 28 patients.

In our study, we included in the study a larger number of patients than the minimum statistically calculated number to increase the likelihood of achieving reliable results.

  1. Notably, only a subgroup of patients (n=24) received 12 months of dienogest. The selection criteria for these patients are not specified in the methods, potentially introducing bias. Was randomization employed? The analysis is paired pre- and post-therapy. Why wasn't a comparison made between patients treated for 1 year and those untreated for 1 year? Clarifications are needed here.

Response 2. Thank you very much for this valuable comment.

All patients from the study group received therapy with progestagens, namely, dienogest 2 mg daily, as recommended in the instruction for the drug for patients over 12 years old. According to clinical recommendations, long-term hormonal therapy is required after surgery to reduce the risk of progression and relapse of the disease, to improve the quality of life, as well as to prevent further complications, including repeated surgical interventions, especially in young patients. We always prescribe hormonal therapy to patients when peritoneal endometriosis is confirmed by laparoscopy in adolescents precisely for these reasons.

The inclusion of 24 patients in our study at the second point of analysis was based on the possibility of readmission to the hospital again after 1 year of the treatment (where we can produce our research specific methods). Since it is very difficult to accumulate patients with endometriosis at a young age due to the prevalence of the disease and delayed diagnosis for an average of 7-10 years, patients were invited for examination from all regions of Russia, which made it possible to collect this group of girls during several years of the research. Some of these patients are still continuing to take medications and we will present the entire sample after some time, when all the girls complete 12 months of treatment; on the other hand, patients from some distinct regions will not be able to fly to Moscow for a re-examination, also from an economic point of view. They were able to fly to the hospital (from Vladivostok region for example) in specifically for surgical intervention, which is covered by the state program, but it’s not necessary for them to have re-examination during treatment in the hospital because they can get recommendations with on-line communication and from a doctor at their place of residence.

SPECIFIC COMMENTS

  1. Incorporate CD206, CD163 (and CD86) into the introduction rather than the discussion.

Response: Dear reviewer, thank you very much for your comment. We kept these markers and their characteristics in the discussion precisely from the position of discussing our results cause-and-effect relationships with this disease at an early age.

  1. Clearly state the study's aim in the body of text (as done in the abstract). Include a hypothesis and rationale, articulating, "In this study, we address... ".

Response: Thank you so much, we clarified in the body of the text:

Page 2 lines 66-68: In this study we address subpopulation dynamics of monocytes and lymphocytes in peripheral blood and peritoneal fluid of adolescents with PE at diagnosis and after 1-year progestogen therapy.

  1. Include a reference to the revised American Society for Reproductive Medicine (rASRM) criteria, specifying the date of revision.

Response: Thank you, we added references in the text.

Page 3 lines 109-112. The laparoscopic report for each case included surgical diagnosis, stage of endometriosis according to the revised American Society for Reproductive Medicine (rASRM classification, revised in 2011) criteria and the revised Enzian Classification (#Enzian, revised in 2021), and description of the lesions in terms of localization, color, size and depth [24–25].

  1. Specify the temperature and time for fixing cell pellets with 2% paraformaldehyde. Explain the choice of 2% PAF for blood isolation (page 3) and 4% PAF for peritoneal fluid isolation (page 4).

Response: Thank you for your comment. Paraformaldehyde fixation protocol: Add PFA to 2% and then incubate at room temperature for 10 minutes, centrifugate tube at 300g, 4° C 5 min. The supernatant was removed and the pellet was resuspended in 10 ml PBS. After second centrifugation the pellet resus-pended in paraformaldehyde 2%.

Page 4 (lines 148-149) mentions a 4% paraformaldehyde solution which, when adding 100 µl sample + 100 µl paraformaldehyde, reaches a concentration of 2%, which is the generally accepted working concentration. 

  1. Address the issue of testing numerous parameters and computing p-values for each. Were the p-values corrected for multiple testing (q-value)?

Response: Thank you very much for your important note. Indeed, we examined a sufficient number of tests for the null hypothesis, however, no more than 10-15 variables, not like for the research analysis in machine learning and genomic data. Besides, we used the False Discovery Rate controlling and Bonferroni amendments.

 To clarify the statistical methods, we have restructured the statistical description:

Lines 185-211 Page 4-5: Statistical data analysis was performed using Statistica 12 software (StatSoftInc., USA) and IBM SPSS Statistics 27. Categorical variables were assessed by calculating frequencies and proportions (%) to compare differences, contingency tables were used and chi-square (χ²) was calculated. For small samples, Fisher's exact test or χ²-Pearson test with Yates continuity correction was determined. Comparison of multiple frequencies was carried out by calculating the χ²-Pearson test. McNemar's chi-square was used to analyze frequency differences in the dependent sample before and after treatment.

The distribution normalities were challenged with distribution histograms visually and indicators of asymmetry, kurtosis, Kolmogorov-Smirnov and Shapiro-Wilk criteria. In addition to checking the type of distribution of variables, the homogeneity of variances in the study groups was assessed using analysis of variance methods, Levene and Brown-Forsyth tests. Comparison of variables with a normal distribution in two groups was performed by parametric Student’s t-test, and a mean value (M) and a standard deviation (SD) were calculated. In multiple groups the analysis of variance (ANOVA) was used. Non-normally distributed variables were described by median (Me) and interquartile range values and compared using Mann–Whitney U-test. Comparison of nonparametric variables in multiple groups was performed using Kruskal-Wallis rank tests. Subsequently, the intergroup differences were assessed by post-hoc test with ranking according to the Dunn or Siegel-Castellan criterion. The adjusted odds ratio (OR) with a 95% confidence interval (CI) were evaluated using logistic regression methods to analyze the influence of various risk factors. The variables in the dependent samples were estimated using Wilcoxon signed rank test and χ2 McNemar’s test for dependent proportions; the trends were considered significant at p<0.05. The results are presented as median, 25-75% quartiles, min-max. Correlations were estimated using Pearson’s correlation coefficient (for normally distributed data) or Spearman’s rank correlation method (nonparametric). The influences of categorical factors and quantitative variables were estimated by factorial ANOVA and multiple logistic regression methods, respectively.

  1. Elaborate on the inclusion of parameters carbohydrate antigen 125 (CA-125) and CA-19.9, explaining their significance in endometriosis management.

Response: We included in the analysis various tumor markers and indicators of inflammation with an assessment of their dynamics during treatment, taking into account that these tumor markers can increase, including against the background of inflammation. Significant dynamics in reducing Ca-125 levels, despite the lack of diagnostic significance of this tumor marker, is important in terms of predicting the effectiveness of therapy not only when monitoring ovarian endometriomas, but also peritoneal endometriosis. The absence of pronounced dynamics in the tumor marker Ca-19.9, which is produced primarily by cells of the gastrointestinal tract, emphasizes the possible specificity of Ca-125 for dynamic assessment in adolescents during treatment.

At the same time, similar articles show rather high potential accuracy of combination of these tumor markers in endometriosis diagnosis in adults, for example:

  1. Chen, J. L. Wei, T. Leng, F. Gao, & S. yu Hou, The diagnostic value of the combination of hemoglobin, CA19-9, CA125, and HE4 in endometriosis. Journal of Clinical Laboratory Analysis, 35 (2021). https://doi.org/10.1002/jcla.23947.

Though, the results in our samples revealed no diagnostic significance for these tests in adolescents.

  1. Refrain from presenting non-significant data as "lower" or indicating trends without statistical significance to prevent potential misinterpretation of data.

Response: Thank you for this fair note. We corrected statistically insignificant results without assessing trends in the text of the article.

Page 6 lines 262-263: Noteworthy, the percentage of monocytes positive for the anti-inflammatory marker CD206 in peritoneal fluid didn’t differ significantly between groups.

  1. Consider adapting Figure 1 into violin plots, especially for CD206, and enhance Figure 3.

Response: Thank you for your valuable suggestion. We consider this blot presentations of figure 1 and 3 as more visually understandable. If it is necessary, we will, of course, eagerly rearrange these figures into violin plots as you kindly recommend us. However, it seems to us that the presentation in the form of violin graphs is less familiar to most readers. Presenting boxplots with whiskers makes it easier to compare similar results between articles of other authors, so for now we would prefer to leave these plots in their original form if you benignly let us.

  1. Improve the presentation of Figure 4, look for consistency.

Response: Thank you for your comment. We presented Figure 4 to demonstrate the factor analysis of the dependence of the clinical effectiveness of hormonal therapy on the quantitative and qualitative composition of the immunophenotypes of lymphocytes and monocytes. We specified the consistency in the title of the picture 4:

Lines 362-369 Page 12. The influence of therapy effectiveness factors on the level of monocytes and lymphocytes. 1): increase in CD45+ and decrease in CD206+ monocytes in patients with amenorrhea after 12 months of therapy. 2): Higher level of cytotoxic CD56dimCD16bright-NK cells with persistent pain and 3) the need to take NSAIDs to relieve pain in patients after 12 months of therapy. 4): In case of pain persistence in patients with endometriosis after 12 months of therapy, the level of CD56brightCD16dim-NK cells with immunoregulatory properties in the blood appeared to be higher. 5): Unidirectional dynamics in blood levels of CD56brightCD16dim and CD56dimCD16bright-NK cells in case of pain persistence after 12 months of therapy.

  1. In Table 2, provide reference values in the left column for clarity.

Response: Thank you for your recommendation. We included reference values in the left column to the table 2.  

  1. Provide a reference for the statement "NK cells known to produce cytokines with immunoregulatory properties."

Response: Thank you for your comment. We added references in the text:

Fu B., Tian Z., Wei H. Subsets of human natural killer cells and their regulatory effects. Immunology.2014; 141:483–9. doi:10. 1111/imm.12224

Moretta, A. What is a natural killer cell? / A. Moretta, C. Bottino, M. C. Mingariet al. // Nature Immunology. – 2002. – Vol. 3, N 1. – P. 6-8.doi: 10.1038/ni0102-6.

Maas-Bauer K., Lohmeyer J. K., Negrin R. S. et al. Invariant natural killer T-cell subsets have diverse graft-versus-host-disease–preventing and antitumor effects/ Blood (2021), 10.1182/blood.2021010887

Michel T.,  Poli A.,  Cuapio A.,  Briquemont B.,  Iserentant G,  Ollert M., Zimmer J. Human CD56bright NK Cells: An Update. J Immunol. 2016 Apr 1;196(7):2923-31. doi: 10.4049/jimmunol.1502570

  1. Specify which metalloproteinases (MMP).

Response: Thank you for your recommendation, we specified:

Lines 424: MMPs -2, -3, -7, -9, -11

  1. Add references for the paragraphs on CD86 (page 14) and CD80 (page 16).

Response: Thank you for your comment. The references were added.

Another studied marker, CD86, significantly increased in the blood of patients with endometriosis, is constitutively expressed on Langerhans cells, memory B cells, macrophages and monocytes. CD86 expression increases rapidly following activation by Ig receptor cross-linking or by addition of certain cytokines, for example, IFN-γ stimulation. Increased counts of CD86+ cells in patients with endometriosis, especially accompanied chronic pelvic pain and persistent dysmenorrhea, indicates extensive inflammatory process and systemic immune response in PE [44–46].

  1. J. Vallvé-Juanico, S. Houshdaran, & L. C. Giudice, The endometrial immune environment of women with endometriosis. Human Reproduction Update, 25 (2019). https://doi.org/10.1093/humupd/dmz018.
  2. Y. Bian, D. L. Walter, & C. Zhang, Efficiency of Interferon-γ in Activating Dendritic Cells and Its Potential Synergy with Toll-like Receptor Agonists. Viruses, 15 (2023). https://doi.org/10.3390/v15051198.
  3. T. Shiraishi, M. Ikeda, T. Watanabe, Y. Negishi, G. Ichikawa, H. Kaseki, S. Akira, R. Morita, & S. Suzuki, Downregulation of pattern recognition receptors on macrophages involved in aggravation of endometriosis. American Journal of Reproductive Immunology, 91 (2024). https://doi.org/10.1111/aji.13812.

We identified a trend towards increased percentage of CD80+ cells in PE, apparently linked to the inflammatory component of the disease, albeit below the threshold of significance [57–58].

  1. L. A. Horn, T. M. Long, R. Atkinson, V. Clements, & S. Ostrand-Rosenberg, Soluble CD80 protein delays tumor growth and promotes tumor-infiltrating lymphocytes. Cancer Immunology Research, 6 (2018). https://doi.org/10.1158/2326-6066.CIR-17-0026.
  2. J. Olkowska-Truchanowicz, A. Białoszewska, A. Zwierzchowska, A. Sztokfisz-Ignasiak, I. Janiuk, F. Dabrowski, G. Korczak-Kowalska, E. Barcz, K. Bocian, & J. Malejczyk, Peritoneal fluid from patients with ovarian endometriosis displays immunosuppressive potential and stimulates th2 response. International Journal of Molecular Sciences, 22 (2021). https://doi.org/10.3390/ijms22158134.

Comments on the Quality of English Language

PRESENTATION AND LANGUAGE

Comments:

Consistently present p-values as p=0.xxx in the abstract and text. Specify which markers indicate M1/M2 polarization.

Response: Thank you for this recommendation. We changed in the abstract:

Page 1 Lines 27-28: During the treatment, cytotoxic lymphocytes CD56dimCD16bright (p=0.049) and CD206+ monocytes (p<0.001) significantly increased while CD163+ monocytes decreased in number (p=0.017).

Thank you very much for your recommendation. We did not specifically dwell on the paradigm of M1-M2 polarization of macrophages, since nowadays it is recognized as rather conditional and changeable. Therefore, we specified commonly M1/M2 polarization markers in Methodology section:

Page 3 Lines 138-139: The monocyte markers included conditionally M2 anti-inflammatory (CD206, CD163) and M1 pro-inflammatory (CD86, CD80) characteristics of monocytes and macrophages.

Improve English grammar, correct typos, and translate "Железо" to English.

Response: Thank you for this correction, we tried to improve English grammar throughout the document. We corrected and translated “Железо” to Total Iron (Fe2+) in the table 2.

Maintain consistency in using exponent notation (e.g., CD56brightCD16-/dim) throughout the manuscript for readability.

Response: Thank you for this suggestion, we have corrected throughout the text to avoid exponent notation.

Include units for general blood test parameters, biochemical, and hormonal profiles.

Response: Thank you for this suggestion, we have corrected and included units in the body of the text:

Page 5 Lines 238-247: Comparison of general blood test parameters, biochemical and hormonal profiles, blood coagulation system parameters did not reveal differences between the groups, with the exception of a higher level of prolactin in patients with endometriosis compared to the comparison group (468.21±303.23 mlU/l vs. 276.52±125.91 mlU/l, p= 0.041) and lower values of platelets (276.89±64.14 109 per l vs. 313.82±85.14 109 per l, p=0.042) and thrombocrit (0.27±0.05 l/l vs. 0.31±0.07 l/l, p=0.016) in patients with endometriosis. It is important to note that in adolescents the level of AMH in the groups was also comparable (4.24±2.55 ng/ml versus 4.46±2.72 ng/ml, p=0.6). Of indirect markers of inflammation, C-reactive protein (1.03±1.02 mg/l versus 0.81±0.79 mg/l, p=0.4), the level of CA-125 (25.41±25.93 U/ml vs. 15.96±6.10 U/ml, p=0.8), Ca-19.9 (18.82±26.14 U/ml vs. 18.55±17.49 U/ml, p=0.6), HE4 (49.82±9.51 pmol/l vs. 50.75±12.24 pmol/l, p=0.9), fibrinogen ( 2.65±0.51 g/l versus 2.77±0.48 g/l, p=0.4), glucose (4.68±0.50 mmol/l versus 4.78±0.25 mmol/l, p=0.4), total iron in the blood serum (Fe2+) (20.71±9.74 µmol/l versus 20.23±9.38 µmol/l, p=1.0) and ferritin (29.46±21.94 µg/l vs. 28.50±18.56 µg/l, p=0.7) levels did not differ significantly between the two groups.

Use the official name FlowJo, LLC when referencing.

Response: Thank you for this correction, we changed in the text:

Lines 142-143: The flow cytometry data were analyzed using FlowJo LLC v10 software (FlowJoLLC, USA).

Enhance referencing format, e.g., ([7–10], [11], [5–12]) or ([13][12–14–15]).

Response: Thank you for this improvement, we changed throughout the text the reference format.

Comment: Always provide line and page numbers for peer-reviewing documents.

Response: Thank you for this recommendation, we added line numbers in the document for the convenience.

We thank You very much for your suggestions!

Reviewer 2 Report

Comments and Suggestions for Authors

Dear authors,

I read your manuscript with great interest. Endometriosis is a chronic disease with high social impact. As is now known, the pathology has an early onset, so a diagnosis in adolescents is essential to reduce the diagnostic delay. The disease certainly has a hormone-related inflammatory basis and knowing the inflammatory factors that determine it can help us treat it and reduce its possible progression.

My suggestions to improve the manuscript:

-Introduction: add recent references regarding the prevalence of disease in adolescents and young women (doi: 10.1016/j.fertnstert.2020.06.012) (doi: 10.1016/j.fertnstert.2022.12.004) (doi: 10.1016/j.fertnstert.2022.12.039)

- Result: if possible, describe the reduction or persistence of US or MRI features of disease after medical treatment with progestins; if possible describe the possible relation between the modification of monocyte/lymphocyte phenotypes and modification of painful symptoms in patients on hormone therapy 

-Discussion: debating the possible role of your results respect an early noninvasive diagnosis and respect the therapy 

Sincerely

Author Response

Response to Reviewer 2 Comments

Comment. We thank the reviewer for their careful reading of the manuscript and interest in our study. The point-by-point answers to the comments are given below. 

Comment 1. Introduction: add recent references regarding the prevalence of disease in adolescents and young women (doi: 10.1016/j.fertnstert.2020.06.012) (doi: 10.1016/j.fertnstert.2022.12.004) (doi: 10.1016/j.fertnstert.2022.12.039).

Response 1. Thank you for your suggestion. We added recommended references in the introduction.

First symptoms of endometriosis often develop in adolescence, an average 7-10 years before the diagnosis [1–2], [3]. The delayed diagnosis and treatment may cause uncontrolled progression, undermine conservative treatment options and ultimately lead to infertility and impaired quality of life in young patients [4–5], [6]. Despite the prevalence of endometriosis, particularly its peritoneal form (PE), clinical findings concerning initial forms of the disease in adolescents remain limited [7–8], [9].

Comment 2. Result: if possible, describe the reduction or persistence of US or MRI features of disease after medical treatment with progestins; if possible describe the possible relation between the modification of monocyte/lymphocyte phenotypes and modification of painful symptoms in patients on hormone therapy

Response 2. Thank you for this valuable recommendation. The special features of clinical signs of endometriosis in adolescents are indeed of great value. Currently, non-invasive diagnosis of peritoneal endometriosis in adolescents remains a huge problem. In particular, according to transabdominal and transrectal ultrasound data, we can assume the presence of peritoneal endometriosis in an average of 5% of adolescents in this group with already confirmed PE. According to MRI data, in 70-85% of cases we are able to suspect superficial PE in adolescents with a detailed description of the study by experts in the field. We summarized the data comparing instrumental diagnosis of endometriosis in adolescents in a previous article:

Khashchenko, E.P.; Uvarova, E.V.; Fatkhudinov, T.K.; Chuprynin, V.D.; Asaturova, A.V.; Kulabukhova, E.A.; Vysokikh, M.Y.; Allakhverdieva, E.Z.; Alekseeva, M.N.; Adamyan, L.V.; Sukhikh, G.T. Endometriosis in Adolescents: Diagnostics, Clinical and Laparoscopic Features. J. Clin. Med. 2023, 12, 1678. https://doi.org/10.3390/jcm12041678

Given the difficulties in non-invasive diagnostics, we experience the same difficulties in assessing the dynamics of instrumental data during treatment with progestins.  Therefore, first of all, we focus on clinical data on the dynamics of dysmenorrhea, pain intensity according to VAS, quality of life and other manifestations of endometriosis (gastrointestinal and dysuria). These data we also summarized in previous article:

Khashchenko, E.P.; Uvarova, E.V.; Chuprynin, V.D.; Pustynnikova, M.Y.; Fatkhudinov, T.K.; Elchaninov, A.V.; Gardanova, Z.R.; Ivanets, T.Y.; Vysokikh, M.Y.; Sukhikh, G.T. Pelvic Pain, Mental Health and Quality of Life in Adolescents with Endometriosis after Surgery and Dienogest Treatment. J. Clin. Med. 2023, 12, 2400. https://doi.org/10.3390/jcm12062400

However, despite the reduction in clinical symptoms and improvement in quality of life, according to molecular studies, we see that the indicators did not normalize in adolescent patients after a year of therapy. In those patients whose clinical situation required diagnostic liquid non-contact hysteroscopy, we saw persistence of adenomyosis a year later during therapy.

Therefore, our goal at the moment was to assess the dynamics of these immunological characteristics after a year of the therapy and justify the necessity of prolonged therapy. Further, we plan to continue the study and evaluate the same indicators after another year of therapy and identify prognostic markers of the clinical effectiveness and the risk of relapse after treatment discontinuation.

The relation of the monocytes/lymphocytes phenotypes and painful symptoms on hormonal therapy we tried to summarize in the picture 5. In adolescent patients with PE we observed decreased CD16+monocyte counts which may indicate decreased sensitivity to pro-inflammatory patterns, reduced number of monocytes capable of peripheral tissue colonization and ultimately a decreased anti-lesion protective capacity of the immune system already at initial stages of the disease. Increased counts of CD86+cells in patients with endometriosis, as well as in cases of chronic pelvic pain and persistent dysmenorrhea, indicates extensive inflammatory process and systemic immune response in PE. One-year conservative treatment with progestogens according to our findings promoted an increase in NK cell counts in peripheral blood reflecting probable activation of peritoneal cytotoxic immune response for antiendometriotic purposes. At the same time a year of hormonal therapy was accompanied with continued high anti-inflammatory activity of monocytes, exactly CD+206 monocytes, responsible for vascularization and endometrioid lesions progression. Monocyte counts positive for scavenger receptor CD163 decreased during the treatment, possibly in connection with treatment-related amenorrhea and the absence of retrograde menstruation, indirectly indicating a decrease in oxidative stress levels within the peritoneal cavity.

Comment 3. Discussion: debating the possible role of your results respect an early noninvasive diagnosis and respect the therapy

Response 3. Thank you for this fair suggestion. We previously assessed some range of molecular parameters in adolescents with endometriosis not only with evaluation of immunological markers, but also according to mitochondrial functioning and biogenesis, as well as apoptosis and autophagy:

Khashchenko, E.P.; Vysokikh, M.Y.; Marey, M.V.; Sidorova, K.O.; Manukhova, L.A.; Shkavro, N.N.; Uvarova, E.V.; Chuprynin, V.D.; Fatkhudinov, T.K.; Adamyan, L.V.; et al. Altered Glycolysis, Mitochondrial Biogenesis, Autophagy and Apoptosis in Peritoneal Endometriosis in Adolescents. Int. J. Mol. Sci. 2024, 25, 4238. https://doi.org/10.3390/ijms25084238

We actually recalculated diagnostic significance for each marker and immunological aspects also, however, the significance was inferior to pain assessment scale using VAS in combination with MRI criteria or indicators of pain chronicity and persistence of dysmenorrhea. As for our results of monocytes and lymphocytes levels, the most perspective results we assessed for the percentage of cytotoxic CD56dimCD16bright NK cells both before treatment (OR 5.25, DI 0.44–62.12, p<0.001) and after treatment (OR 1.92, DI 0.94–3.93, p=0.006) as significant predictors of the pain relief efficacy after a year of treatment (logistic regression analysis) (page 13). Also we observed treatment-related dynamics of CD206+ monocytes in the blood which was associated with pain relief efficacy (OR 0.82, DI 0.62–1.08, Chi2=4.29, p=0.038), playing an essential role in cell metabolic reprogramming and transition to glycolysis as the dominant energy production mode, concomitant acidosis, and angiogenesis in the foci.

We thank You very much for your suggestions!

Round 2

Reviewer 1 Report

Comments and Suggestions for Authors

The authors have done significant improvement on their paper. The MS has been addressed according to all previous comments, and it is (almost) suitable for sharing with the scientific community. 

The reviewer requests only presentation change:
Please amend for misplaced data to Figure 3 at P9, L317 (CD45, before/after treatment). Please also include in the legend. 
Please also correct typos.

Comments on the Quality of English Language

correct spelling "Brown–Forsythe test"

delete "Место для вводse".

Author Response

Response to Reviewer 1 Comments

Comment. We thank the reviewer for their interest in our study and careful reading of the manuscript. The point-by-point answers to the comments are given below. 

Comment: The reviewer requests only presentation change:

Please amend for misplaced data to Figure 3 at P9, L317 (CD45, before/after treatment). Please also include in the legend.

Please also correct typos.

Response:

Thank you so much for correction. We changed in the legend the description to “increase in mononuclear cells (CD45+)”, as CD45 is a pan-leukocyte marker, used for gating.

Page 12, figure 4, line 370

1): increase in mononuclear cells (CD45+) and decrease in CD206+ monocytes in patients with amenorrhea after 12 months of therapy

Comments on the Quality of English Language correct spelling "Brown–Forsythe test"

Response:

Thank you very much, we corrected Brown–Forsythe test (line 201).

delete "Место для вводse".

Response:

Thank you very much, we have deleted and replaced with reference [41]:

Page 15, line 489-490

Increased counts of CD86+ cells in patients with endometriosis, especially accompanied chronic pelvic pain and persistent dysmenorrhea, indicates extensive inflammatory process and systemic immune response [41].

Shiraishi, T., Ikeda, M., Watanabe, T., Negishi, Y., Ichikawa, G., Kaseki, H., … Suzuki, S. (2024). Downregulation of pattern recognition receptors on macrophages involved in aggravation of endometriosis. American Journal of Reproductive Immunology91(1). https://doi.org/10.1111/aji.13812

We thank You very much for your suggestions, positive feedback and time to improve our manuscipt!